# Mechanism of small molecule inhibition of *Plasmodium falciparum* myosin A informs antimalarial drug design

Dihia Moussaoui[1,8,9], James P. Robblee[2,9], Julien Robert-Paganin [1,9], Daniel Auguin [1,3], Fabio Fisher[4], Patricia M. Fagnant[2], Jill E. Macfarlane[2], Julia Schaletzky [5], Eddie Wehri[5], Christoph Mueller-Dieckmann[6], Jake Baum [4,7], Kathleen M. Trybus [2] ✉ & Anne Houdusse [1] ✉

Malaria results in more than 500,000 deaths per year and the causative *Plasmodium* parasites continue to develop resistance to all known agents, including different antimalarial combinations. The class XIV myosin motor PfMyoA is part of a core macromolecular complex called the glideosome, essential for *Plasmodium* parasite mobility and therefore an attractive drug target. Here, we characterize the interaction of a small molecule (KNX-002) with PfMyoA. KNX-002 inhibits PfMyoA ATPase activity in vitro and blocks asexual blood stage growth of merozoites, one of three motile *Plasmodium* life-cycle stages. Combining biochemical assays and X-ray crystallography, we demonstrate that KNX-002 inhibits PfMyoA using a previously undescribed binding mode, sequestering it in a post-rigor state detached from actin. KNX-002 binding prevents efficient ATP hydrolysis and priming of the lever arm, thus inhibiting motor activity. This small-molecule inhibitor of PfMyoA paves the way for the development of alternative antimalarial treatments.

Malaria infection in humans, caused by unicellular parasites from the genus *Plasmodium* and transmitted via the bite of an infected *Anopheles* mosquito, is a major global health challenge[1,2]. In 2020, malaria was responsible for 627,000 deaths, the majority being children under the age of 5 years[3]. Despite the remarkable progress of research aimed at advancing antimalarial therapeutics, the parasite continues to develop resistance to all existing treatments, including the gold-standard front line artemisinin-based combination therapies (ACTs)[4,5]. The recent licensure of the malaria vaccine marks a significant milestone in efforts to control malaria, but the relatively modest efficacy of the RTS,S

vaccine means that complementary approaches will be essential if the WHO's goal of a 90% reduction in rates by 2030 is to be realized[6].

Malaria parasites are motile throughout their complex human and mosquito life cycle. They move by a process called gliding motility[7], which underpins their ability to reach, cross, and enter host tissues and cells. Gliding is powered by a macromolecular complex called the glideosome, the core of which is comprised of a class XIV myosin A (PfMyoA) interacting with short, oriented filaments of a divergent actin (PfAct1)[7,8]. Importantly, PfMyoA has been demonstrated to be essential for parasite motility and for pathogenesis[9–12], thus making this myosin

[1]Structural Motility, Institut Curie, Université Paris Sciences et Lettres, Sorbonne Université, CNRS UMR144, 75248 Paris, France. [2]Department of Molecular Physiology & Biophysics, University of Vermont, Burlington, VT, USA. [3]Laboratoire de Biologie des Ligneux et des Grandes Cultures (LBLGC), Université d'Orléans, INRAE, USC1328 Orléans, France. [4]Department of Life Sciences, Imperial College London, Exhibition Road, South Kensington, London SW7 2AZ, UK. [5]Center for Emerging and Neglected Diseases, Drug Discovery Center, Berkeley, CA, USA. [6]Structural Biology group, European Synchrotron Radiation Facility (ESRF), 71, Avenue des Martyrs, 38000 Grenoble, France. [7]School of Medical Sciences, Faculty of Medicine & Health, UNSW Sydney, Kensington, NSW 2052, Australia. [8]Present address: Structural Biology group, European Synchrotron Radiation Facility (ESRF), 71, Avenue des Martyrs, 38000 Grenoble, France. [9]These authors contributed equally: Dihia Moussaoui, James P. Robblee, Julien Robert-Paganin. ✉e-mail: kathleen.trybus@uvm.edu; anne.houdusse@curie.fr

motor a desirable target for preventing lifecycle progression and as such malaria disease.

The development of small molecules able to specifically activate or inhibit myosin force production has been successful in several other myosin classes, including the activator omecamtiv mecarbil (OM) and the inhibitor mavacamten (CAMZYOS™), targeting β-cardiac myosin against heart failure and inherited cardiac diseases[13–15] and CK-571, a smooth muscle myosin 2 inhibitor (SmMyo2) against asthma[16]; reviewed by[17–19]. From Structure Activity Relationship (SAR) on the Blebbistatin (Blebb) scaffold, MPH-220 was developed as a specific inhibitor of skeletal muscle myosin (SkMyo2) with promising therapeutical value against muscular spasticity[20]. The fact that some of these compounds such as OM and mavacamten have completed phase 3 clinical trials[21,22] validates the development of myosins as realistic and druggable targets. Mavacamten (CAMZYOS™) has recently been approved by the FDA for the treatment of obstructive hypertrophic cardiomyopathy to improve functional capacity and symptoms.

Here, we characterize the interaction of PfMyoA with KNX-002, a small molecule inhibitor of PfMyoA ATPase activity and of *P. falciparum* merozoite asexual blood stage growth. KNX-002 was initially identified as an inhibitor from high-throughput actin-activated screens performed by Cytokinetics, Inc. The X-ray structure of PfMyoA complexed to KNX-002 identifies the binding pocket, and combined with transient kinetics and binding studies reveal how the compound sequesters PfMyoA in a state of low affinity for actin. KNX-002 is thus a promising candidate to develop an antimalarial treatment based on inhibiting the myosin motor that powers motile stages of the parasite, which are essential for infection.

## Results

### KNX-002 is a promising small molecule antimalarial inhibitor

KNX-002 was initially identified as an inhibitor of PfMyoA from high throughput actin-activated ATPase screens performed by Cytokinetics, Inc. using 50,000 compounds from their library. The same compounds were screened against TgMyoA in parallel, and upon completion of the screen KNX-002 was identified as a robust inhibitor of both class XIV myosins[23] (see Methods for screen details). KNX-002 inhibits both the actin activated (IC$_{50}$ = 7.2 μM) and the basal (IC$_{50}$ = 3.6 μM) ATPase activity of PfMyoA, while showing little effect on cardiac or skeletal

myosin II (Fig. 1a, b). Measurement of the affinity of KNX-002 for nucleotide-free PfMyoA (8.6 μM) and PfMyoA.ADP (11.1 μM) yielded values (Supplementary Fig. 1a, b) that were of the same order of magnitude as the IC$_{50}$ obtained from the steady-state ATPase activities. KNX-002 also inhibited the ability of PfMyoA to move actin in an in vitro motility assay. In the absence of KNX-002 robust motility of actin filaments was observed at speeds of 3.33 ± 0.50 μm/s (Supplementary Movie 1), while there was no directed actin filament motion in the presence of 200 μM KNX-002 (Supplementary Movie 2). In addition, very few actin filaments were bound to surface-immobilized myosin in the presence of KNX-002 (6.5 ± 1.9 filaments/field versus 110.3 ± 1.9 filaments/field in its absence, n = 6 fields). In vitro TIRF polymerization assays further showed that KNX-002 did not affect PfAct1 filament assembly, which polymerized to similar lengths and at the same rate in the absence (16.6 ± 2.3 subunits/s, n = 36) or presence (17.2 ± 2.5 subunits/s; p = 0.28) of KNX-002.

Because PfMyoA is essential for blood cell invasion[9,10], we tested the effect of KNX-002 on *P. falciparum* asexual, blood-stage growth, an assay dependent on the ability of merozoites to invade erythrocytes[24]. KNX-002 inhibited asexual blood stage growth of merozoites (IC$_{50}$ = 18.2 μM), confirming a drug effect on parasite cells ex vivo (Fig. 1c). Taken together, these results demonstrate that KNX-002 is an inhibitor of PfMyoA activity in vitro, as well as impeding asexual parasite blood stage growth that is a measure of the ability of parasites to invade, replicate, exit and reinvade erythrocytes, resulting in the symptomatic stage of malaria. The observation that KNX-002 inhibits actin-activated ATPase activity and parasite blood stage growth indicates that the compound inhibits myosin when physiological concentrations of nucleotide are present.

### KNX-002 targets the post-rigor state

To dissect the inhibition mechanism of KNX-002 on the force production cycle, we crystallized full-length PfMyoA complexed with KNX-002. We successfully solved the structure of full-length PfMyoA complexed to both the ATP analog Mg.ATP-gamma-S (Mg.ATPγS) and KNX-002 (PfMyoA/ATPγS/KNX-002) at a resolution of 2.2 Å. The structure of full-length PfMyoA complexed to Mg.ATPγS without the compound (PfMyoA/ATPγS/Apo) was also solved at a resolution of 2.03 Å (Supplementary Table 1).

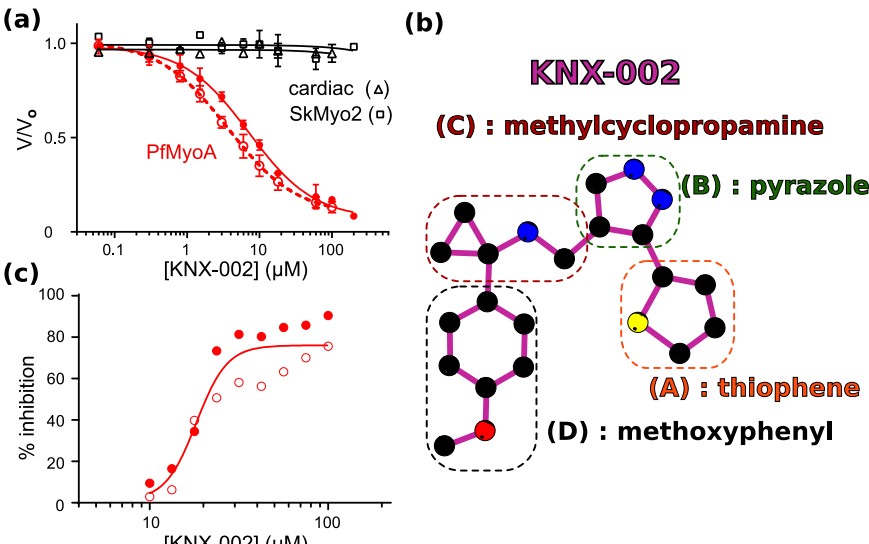

**Fig. 1 | KNX-002 inhibits PfMyoA. a** KNX-002 inhibits actin-activated (filled red circles, IC$_{50}$ = 7.2 μM, (95% CI, 5.8−9.0 μM), n = 3) and basal ATPase activity (open red circles, IC$_{50}$ = 3.6 μM (95% CI, 3.0−4.5 μM), n = 3). KNX-002 has little effect on the actin-activated ATPase of skeletal myosin (SkMyo2, black squares, IC$_{50}$ > 200 μM, n = 2) or cardiac myosin (black triangles, IC$_{50}$ > 100 μM, n = 2).

PfMyoA data are represented as mean values ± SD. **b** The constituent groups of the KNX-002 structure are indicated. **c** The inhibition of asexual parasite blood stage growth by KNX-002 was quantified (IC$_{50}$ = 18.2 μM, n = 2). Source data are provided as a Source Data file.

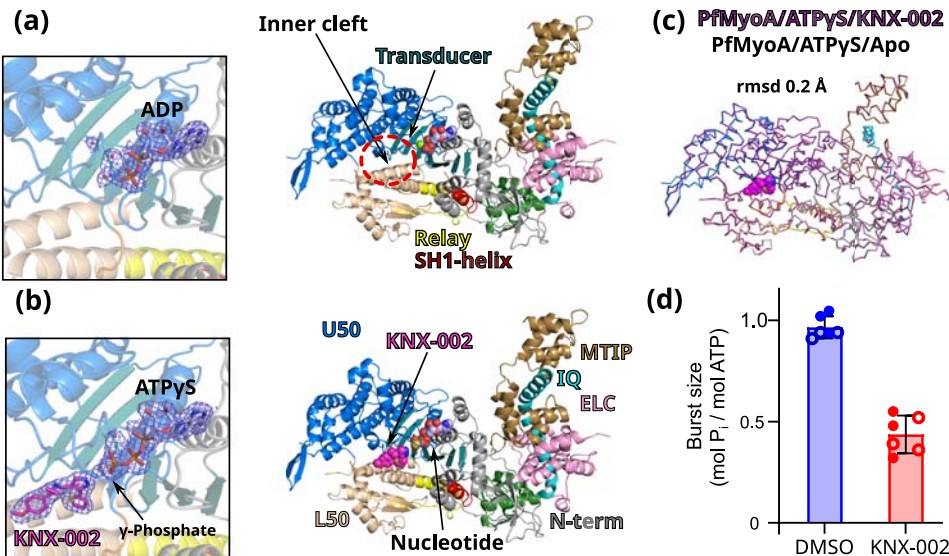

**Fig. 2 | KNX-002 targets the post-rigor (PR) state. a** Structure of PfMyoA in the apo condition (PfMyoA/ATPγS/Apo). Only ADP is found in the 2Fo-Fc electron density map contoured at 1.0 σ (on the left). ATPγS was thus hydrolyzed by myosin, in contrast to the same experiment performed in the presence of KNX-002. **b** Structure of PfMyoA complexed with KNX-002 and MgATPγS (PfMyoA/ATPγS/KNX-002). The compound, MgATPγS, the water molecules and the Mg$^{2+}$ ion can be clearly identified in the 2Fo-Fc electron density map contoured at 1.0 σ (on the left). The gamma-phosphate of ATPγS is present in the density. U50, Upper 50 kDa subdomain; L50, Lower 50 kDa subdomain. **c** The compound does not induce major structural rearrangements upon binding. PfMyoA/ATPγS/KNX-002 and

PfMyoA/ATPγS/Apo are both in a PR state and superimpose quite well with a rmsd of 0.2 Å using the Cα atoms. Zoom on the regions with maximum differences between the two structures show local displacements of side chains (see Supplementary Fig. 3a). **d** Manual quenching experiments show a decreased phosphate burst from 0.97 ± 0.05 mol $P_i$/mol ATP in the absence of compound to 0.44 ± 0.08 mol $P_i$/mol ATP in the presence of 100 μM KNX-002 ($p < 0.0001$, two-tailed t-test with Welch's correction). Data represent two experiments each performed in triplicate with independent protein preparations (open and filled circles). Data bars are mean ± SD. Source data are provided as a Source Data file.

These experiments provide a direct comparison of the apo and the KNX-002-bound structures when a hydrolysable ATP analog is bound. The two datasets are both at high resolution and the resulting electron density maps allow us to determine that the PfMyoA structure adopts a post-rigor (PR) state, an ATP-bound myosin structural state with low affinity for the actin track that is populated upon detachment of the motor from actin prior to the priming of its lever arm (Supplementary Fig. 2). These high-resolution datasets also permitted us to position the nucleotides and KNX-002 without ambiguity, as well as the water molecules, in particular those in the active site (Fig. 2a, b). The two structures are highly superimposable (rmsd 0.2 Å on 914 Cα-atoms, Fig. 2c), indicating that KNX-002 does not induce major structural changes in the myosin structure. Some limited adjustments occur for the residues around the pocket upon KNX-002 binding (max difference in Cα positions being <2.5 Å, Supplementary Fig. 3a). The compound is close to the active site and interacts with Switch-2, an important connector of the motor that changes its conformation during the recovery stroke to favor ATP hydrolysis (Supplementary Fig. 2a, b).

The major difference between the apo and KNX-002-bound structures is the state of the nucleotide. While unhydrolyzed Mg.ATPγS occupies the active site of PfMyoA/ATPγS/KNX-002, the substrate was hydrolyzed in PfMyoA/ATPγS/Apo leaving only Mg.ADP in the active site (Fig. 2a, b). Because cleavage was only observed in the absence of compound, this result implies that KNX-002 directly inhibits the hydrolysis process.

To test this hypothesis, we performed manual quench experiments. In the presence of KNX-002, the phosphate burst is reduced 2.2-fold (Fig. 2d), which demonstrates the ability of KNX-002 to directly reduce ATP hydrolysis. In the motor cycle, ATP hydrolysis occurs after the recovery stroke. This ATP-bound transition is essential for lever arm priming and leads to the pre-powerstroke state (PPS) in which nucleotide-binding loops, notably Switch-2, are positioned to favor ATP hydrolysis (Supplementary Fig. 2). Our

results suggest that KNX-002 inhibits hydrolysis by stabilizing the post-rigor state, thus blocking the recovery stroke. This agrees with our observation that attempts at crystallizing the motor in the presence of KNX-002 when the motor adopts the pre-powerstroke state (see Methods) lead to crystals lacking compound. In fact, the conformation of the KNX-002 binding pocket is drastically different in the PPS state and thus becomes incompetent for KNX-002 binding (Supplementary Fig. 3b).

## The KNX-002 binding pocket
In the PfMyoA/ATPγS/KNX-002 structure, KNX-002 is buried in a tight cryptic cleft located between the so-called Upper 50 kDa (U50) and Lower 50 kDa (L50) subdomains of the motor domain, both of which have elements that bind the actin filament. KNX-002 is bound in the "inner cleft" (Figs. 2a and 3a), close to the so-called back-door of myosin[25,26] that is located near the yP$_i$ of the nucleotide (Fig. 3a). Supplementary Fig. 2a illustrates the distinct conformations of the backdoor in the post-rigor and the pre-powerstroke states of the motor cycle. Specific connectors essential in allosteric transduction[26,27] are directly involved in the binding of KNX-002: Switch-1, Switch-2 and Transducer (Fig. 3a). This structure reveals a small-molecule co-crystallized in the PR state while binding near Switch-2. Most myosin inhibitors described to date target the PPS state of the myosin cycle[14,20,28].

Binding of the KNX-002 molecule, comprised of four moieties (Fig. 1b), appears to be stabilized mainly by the hydrophobic effect: the methoxyphenyl (D) and the methylcyclopropamine (C) are sandwiched between apolar groups of residues from the Switch-1, β7 strand of the Transducer and the φLL-linker, a highly conserved linker located after the $_{Transducer}$β7 and on the U50 side (Supplementary Fig. 3c), as well as apolar residues from the $^{L50}$HP- and $^{L50}$HW-helices (Fig. 3a, b, Supplementary Table 2). A polar interaction is established between the carbonyl of Leu272 and the methylcyclopropamine (C). The pyrazole (B) and thiophene moieties (A) are stacked between residues from the

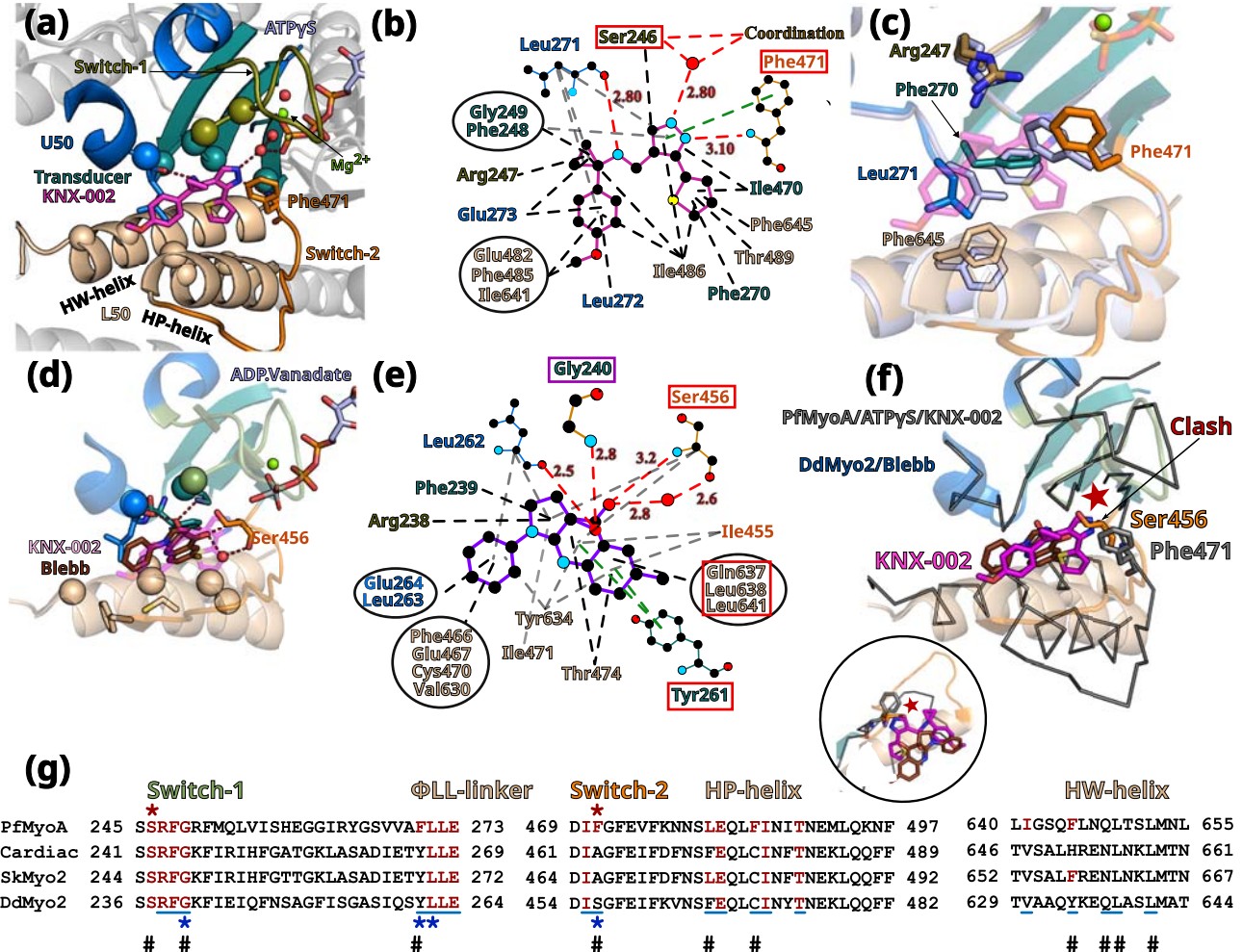

**Fig. 3 | The inner pocket in which KNX-002 binds greatly differs from that of Blebbistatin (Blebb). a** KNX-002 binding pocket. Regions involved in binding are: Switch-2 (orange); Transducer (deep teal cyan); U50 (marine blue); L50 (wheat). Residues are displayed as spheres when involved in apolar interactions; as sticks when involved in electrostatic or π-stacking bonds. **b** Schematic representation of the binding pocket of KNX-002. Each type of interaction is represented differently (Polar interactions, red dashed lines; π-stacking, green; apolar, black). Squares indicate residues involved in different types of bonds for KNX-002 and Blebbistin (shown in **e**). **c** Superimposition of PfMyoA/ATPγS/KNX-002 (colored by sub-domains) and PfMyoA/ATPγS/Apo (gray-blue), both in the post-rigor state. Residues with different conformation (sticks), indicate how adjustments are required to bind KNX-002. **d** Blebb binding pocket in *Dictyostelium discoideum* myosin 2 (DdMyo2, PDB code 1YV3[28]) with the U50 subdomain orientation as in 3a for PfMyoA. KNX-002 (pale purple) does not fit in this pocket as the pre-powerstroke conformation of Switch-2 clashes with the KNX-002 position. Supplementary Table 2 and Supplementary Movie 3 further illustrate how the KNX-002 and Blebb

binding sites differ. **e** Schematic representation of interactions around Blebb. Residues that differ from PfMyoA but bind Blebb are in a red box; conserved residue involved in different bond types (purple box). **f** KNX-002 and Blebb target different pockets. DdMyo2/Blebb (cartoon, colored by subdomain) and PfMyoA/ATPγS/KNX-002 (ribbon, colored in black) are superimposed on the U50 subdomain (residues 182-463 and 604-631). KNX-002 and Blebb binding pockets strongly differ in the conformation of Switch-2 and in the orientation of HP- and HW-helices. Another orientation is represented as a zoom in a circle. **g** Sequence alignment of PfMyoA, β-cardiac MYH7 (cardiac), skeletal muscle myosin 2 (SkMyo2) and DdMyo2 for analysis of the conservation of residues involved in KNX-002 and Blebb binding. Residues involved in KNX-002 binding are colored red in PfMyoA and in other myosins when conserved. Residues involved in Blebb binding are underlined (blue). Residues involved in electrostatic or stacking interactions with KNX-002 (red star), and those involved with Blebb (blue star) are indicated. Remarkable positions that distinguish KNX-002 and Blebb binding modes are marked with a #.

β5 and β6 strands of the Transducer, the $^{L50}$HW-helix with contribution of Switch-2. Amongst these interactions, the most remarkable are established by the pyrazole (B), as it is involved in (i) a polar interaction with a water molecule coordinating the γ-$P_i$ of the nucleotide and (ii) π-stacking electrostatic interactions with $_{Switch-2}$Phe471 (Fig. 3a, b, Supplementary Table 2). The tight fit of KNX-002 in this pocket is remarkable because all four cyclic entities make at least one contact within a radius of ~3.4 Å with atoms of the pocket.

In addition to the small main-chain shifts to adopt the size of the pocket around KNX-002 (Supplementary Fig. 3a), side chains of key-binding residues rotate to fit KNX-002 binding (Fig. 3c). This is specifically true for $_{Switch-2}$Phe471, $_{φLL-linker}$Leu271 and $_{HW-helix}$Phe645 (Fig. 3c).

## The KNX-002 binding mode differs from that described for Blebbistatin

It is important to describe how the Myo2 inhibitor Blebbistatin (Blebb) and KNX-002 binding modes differ because they involve similar PfMyoA structural elements. Indeed, the Blebb binding pocket also involves the Switches-1 and −2, the Transducer and φLL-linker, as well as the HP- and HW-helices (Fig. 3d, e). However, while the binding of KNX-002 and Blebb partially involve the same residues, the pockets conformations are in fact very different (Supplementary Movie 3) and thus KNX-002 cannot fit in the Blebb pocket and vice-versa (Fig. 3f). The two compounds indeed target distinct structural ATP-states of the motor: KNX-002 binds the PR state while Blebb targets the PPS state[20,28] (Supplementary Table 3). Moreover, Blebb and KNX-002 have

different scaffolds and although binding of both mainly involve apolar interactions, the nature of the interactions and the residues involved differ (Fig. 3b, e, g). In addition, the nature of the few polar interactions is drastically different. For example, the hydroxyl moiety of Blebb is involved in essential electrostatic interactions with the amide of $_{Switch-1}$Gly240 and π-stacking interactions are established between Blebb and the $_{φLL-linker}$Y261 (Dd, 1YV3, Fig. 3e, g), while mostly apolar interactions are made with the PfMyoA equivalent $_{Switch-1}$Gly249and $_{φLL-linker}$Phe270 residues and KNX-002. One polar contact occurs similarly in the two pockets with an equivalent carbonyl ($_{φLL-linker}$L271 in PfMyoA and $_{φLL-linker}$L262 in *Dd*Myo2). The Switch-2 conformation as found in the pre-powerstroke state excludes the possibility of a direct bond between Blebb and the nucleotide or the $Mg^{2+}$ ion. In contrast, specific interactions occur for KNX-002 with an ATP molecule bound to the active site when the Switch-2 adopts the conformation found in the post-rigor state. Taken together, these results clearly demonstrate that KNX-002 binds with a previously undescribed binding mode with unique features. An interesting characteristic of the KNX-002 pocket is that it involves a water molecule that directly binds the γ-$P_i$ of the ATP molecule bound in the active site.

Sequence polymorphism between Myo2s and PfMyoA explain the specificity of the two inhibitors for their corresponding binding sites and result in different specific requirement for their binding mode (Supplementary Fig. 4a, b). In particular, KNX-002 specificity for PfMyoA results from major sequence changes compared to Myo2s: $_{Switch-2}$Phe471 in PfMyoA, involved in essential π-stacking interactions with KNX-002, is replaced by a serine or alanine in Myo2s ($_{Switch-2}$Ala463 in β-cardiac myosin); $_{φLL-linker}$Phe270 in PfMyoA is replaced by a Tyrosine ($_{φLL-linker}$Tyr266 in β-cardiac myosin); $_{HP-helix}$Phe485 in PfMyoA is replaced by a cysteine in β-cardiac myosin ($_{HP-helix}$Cys477) and $_{HP-helix}$Leu481 in PfMyoA is replaced by a Phenylalanine in β-cardiac myosin ($_{HP-helix}$Phe473) (Supplementary Fig. 4a, b). The Phe/Leu polymorphism in the HP-helix was reported to be responsible for the high specificity amongst Myo2s for a Blebb derivative of high therapeutical potential (MPH-220)[20]. In addition, the HW-helix residues involved in binding are poorly conserved: $_{HW-helix}$Phe645 in PfMyoA is replaced by a Histidine in β-cardiac myosin ($_{HW-helix}$His651). Such sequence differences allow each inhibitor to interact specifically in its tight pocket.

We validated this site and the role of some residues in KNX-002 binding and specificity with a triple mutant of PfMyoA. Three residues of the PfMyoA sequence were replaced by that of β-cardiac muscle myosin (Phe270Tyr/Phe471Ala/Phe645His) at key-positions for KNX-002 binding (Fig. 3g; Supplementary Fig. 4a, b). As in cardiac myosin (Fig. 1a), this triple mutant became less sensitive to KNX-002 compared with WT, with basal ATPase $IC_{50}$ increasing from 3.6 to 52 μM, and actin-activated ATPase $IC_{50}$ increasing from 7.2 to >100 μM (Supplementary Fig. 4c). The triple mutant showed intact heavy and light chains on an SDS-gel, and its functionality was assessed by an in vitro motility assay where it moved actin filaments with a Gaussian distribution of speeds that was the same in the presence or absence of KNX-002 (Supplementary Fig. 5). This result is consistent with the binding pocket of KNX-002 identified crystallographically, and suggests that some or all of these three key-aromatic positions are essential in the efficient binding and specificity of KNX-002 for PfMyoA. Interestingly, the specificity of KNX-002 for Class XIV myosins is also suggested by a sequence alignment with human unconventional myosins as differences are found for several residues involved in KNX-002 binding, including those that we have been mutated in the triple mutant (Supplementary Fig. 4b).

### Impact of KNX-002 on the nucleotide binding site

Compared to Blebb, KNX-002 is positioned closer to the active site and the water that binds to its pyrazole group also participates in γ$P_i$ coordination and interacts with a water molecule involved in $Mg^{2+}$ coordination (Fig. 4a, b). This suggests that KNX-002 may influence nucleotide binding rates and affinity, in addition to its hydrolysis.

Transient kinetics were thus used to measure the association and dissociation rate constants of mant-ATP and mant-ADP for PfMyoA, nucleotides whose fluorescence is enhanced when bound to PfMyoA. KNX-002 caused a 4.8 and 3.3-fold decrease in the association rate constant for ATP and ADP, respectively, and small increases in the dissociation rate constants (Fig. 4c). The estimated Kd for ATP and ADP in the presence of KNX-002 was 0.7 μM and 3.5 μM, respectively, implying that under physiological nucleotide concentrations, motor with KNX-002 bound would also have nucleotide bound.

The nucleotide binding experiments (Fig. 4c) and KNX-002 binding experiments (Supplementary Fig. 1) indicate that KNX-002 can bind nucleotide-free (NF) myosin in vitro and thus we determined NF PfMyoA structures bound to KNX-002 to investigate why nucleotides bind more slowly (Supplementary Fig. 6, Supplementary Table 1). Interestingly, the KNX-002 binding site is the same in the NF and the MgATPγS structures, suggesting that the compound traps myosin in similar post-rigor states regardless of nucleotide. Small local differences in the active site were found for residues involved in nucleotide binding (Supplementary Fig. 6). Comparison with MgATPγS-bound structures indicates that mobility in the active site would be required to facilitate nucleotide binding. Thus, the presence of KNX-002 in the inner cleft likely slows these rearrangements resulting in slower nucleotide binding. Our structures suggest that occupation of the nucleotide binding site and of the ligand binding site are not mutually exclusive, although KNX-002 likely slows motor domain dynamics, particularly in the active site.

Taken together, these results suggest that PfMyoA adopts a stable PR state when both ATP and KNX-002 are bound, in which the inner cleft cannot close, thus preventing the recovery stroke. The water near the pyrazole, observed in our PfMyoA-MgATPγS structure bound to KNX-002, is not in position to favor attack of the γ-$P_i$ of ATP. The fact that both ATPγS and KNX-002 are compatible in the same structure and their mutual presence does not lead to hydrolysis of ATPγS indicates that KNX-002 prevents hydrolysis, most likely by perturbing the ability to favorably position the water molecule required for hydrolysis. The observation that KNX-002 separates the active site from $_{Switch-2}$Glu474 and slows Switch-2 rearrangements excludes a ATP hydrolysis mechanism that normally occurs in myosins, which requires the repositioning of Switch-2 during the recovery stroke so that the $_{switch-2}$glutamate can promote ATP hydrolysis by coordinating the attacking water molecule[29,30]. Our structures thus provide a mechanistic explanation for how KNX-002 disfavors ATP hydrolysis.

### PfMyoA.ADP has low affinity for F-actin in the presence of KNX-002

To investigate whether KNX-002 affects the stability of PfMyoA actin-bound states, we performed actin pull-down experiments using surface-bound PfMyoA (see Methods, Fig. 5a, Supplementary Figs. 8 and 9) in the presence of either ATP, ADP, or no nucleotide with or without KNX-002.

As expected, myosin binds few filaments in the presence of ATP, and KNX-002 does not change this property (Fig. 5a). In the absence of nucleotide (Nucleotide Free, NF), a similar large number of actin filaments are recruited in the absence (Apo) or presence of KNX-002. The absence of nucleotide allows myosin to bind actin with high affinity in the Rigor state, a state with the actin-binding cleft closed[31] (Supplementary Fig. 2). Note that in this Rigor state, the φLL-linker directly interacts with the HP- and HW-helices and thus the KNX-002 binding site we identified does not exist. The observation that KNX-002 does not change the amount of actin bound to PfMyoA in NF conditions is consistent with the fact that PfMyoA stays bound in the Rigor state with the inner pocket closed, which would prevent KNX-002 binding (Fig. 5b). KNX-002 also does not affect the rate of ATP induced acto-myosin dissociation, further suggesting that the compound is not compatible with the Rigor state (Supplementary Fig. 10) and instead binds a myosin state detached from F-actin. In ADP, fewer actin

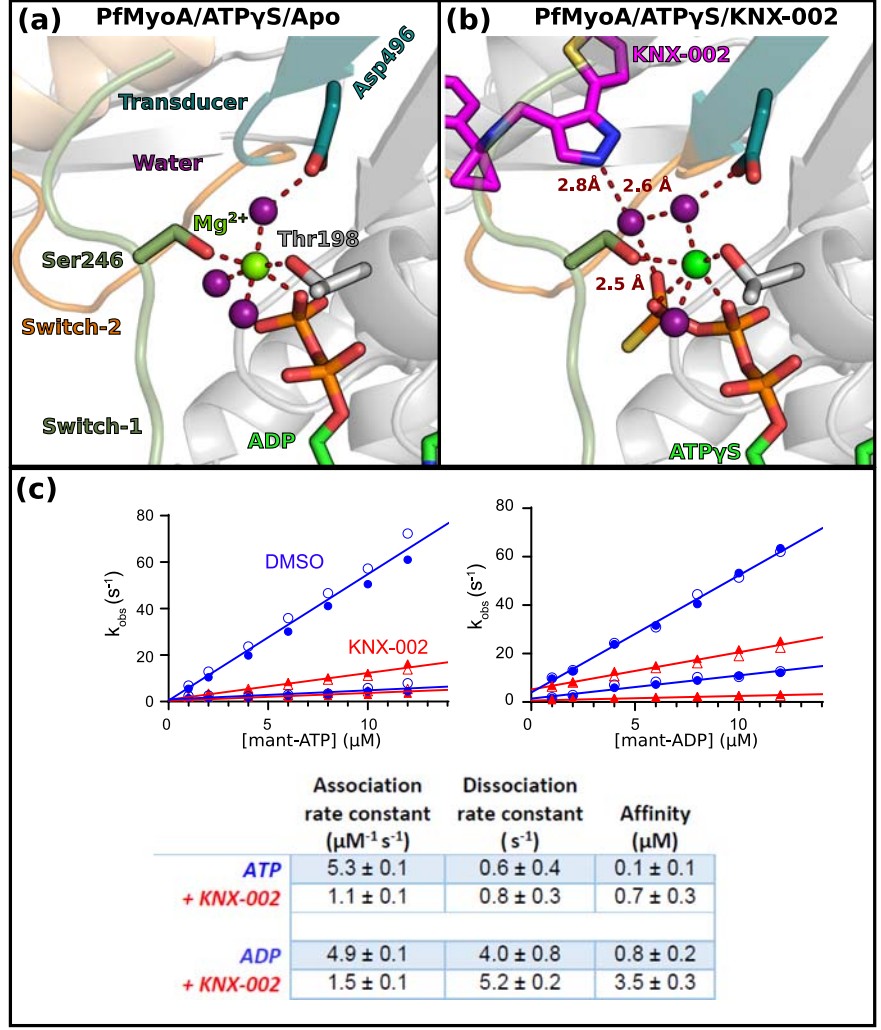

**Fig. 4 | Effects of KNX-002 on the active site.** The $Mg^{2+}$ ion is hexa-coordinated in both the Apo (**a**) and in the KNX-002 bound (**b**) structures. **a** Although ATPγS was used for the crystallization in both structures, the nucleotide is hydrolyzed in the absence of KNX-002 and ADP is found in the active site (**a**). **b** When KNX-002 occupies its pocket, the compound stabilizes a water molecule that also binds the γ-$P_i$ of ATP and a water molecule that coordinates the $Mg^{2+}$ ion. Supplementary Fig. 7 indicates that this additional interaction does not change the hexa-coordination of the $Mg^{2+}$ ion. **c** KNX-002 slows both mant-ATP and mant-ADP binding to PfMyoA. Rates in the absence (blue circles) or presence of 100 μM KNX-002 (red triangles) of the fast and slow phases from the biphasic transients are plotted. The association (slope) and dissociation (y-intercept) rate constants and resulting affinity are given in the Table. Data represent two experiments with independent protein preparations. Source data are provided as a Source Data file.

filaments are recruited compared to the NF condition. More importantly, the presence of both ADP and KNX-002 leads to a level of actin binding comparable to the weak binding observed with ATP (Fig. 5a). This result suggests that PfMyoA.ADP binds KNX-002 and prevents it from binding strongly to the F-actin track. Taken together, these results suggest that KNX-002 favors trapping the motor in a state of poor affinity for the actin track when nucleotide is bound.

Since actomyosin.ADP complexes are in equilibrium with detached M.ADP motors, it is likely that KNX-002 can bind these detached M.ADP motors as they explore conformational states that are close to the post-rigor state. KNX-002 binding would thus stabilize this state and prevent re-attachment to the actin filament in a Strong-ADP bound state. The structures currently available for distinct myosins in the Strong-ADP bound state[26] have indeed shown that the inner cleft is closed in these structures. Thus, KNX-002 binding in the inner cleft would prevent strong binding to the track (Fig. 5b).

## Discussion

KNX-002 inhibits PfMyoA, a myosin shown from knockout experiments to be essential for *Plasmodium falciparum* invasion and motility, and

thus represents a previously untapped target that could be used to prevent *Plasmodium falciparum* pathogenesis[9,10]. The observation that KNX-002 ablates *Plasmodium falciparum* blood stage growth highlights the potential of targeting PfMyoA to develop future antimalarial drugs.

Previous studies of myosin inhibitors have shown that these compounds target binding pockets that are not always present in the apo form[16] and that the accessibility of these cryptic pockets may vary depending on the myosin classes and the structural states targeted[16]. These reasons are why the binding mode of most myosin modulators cannot be predicted. Here we determine the cryptic pocket in which KNX-002 binds in PfMyoA via structural and biochemical data and validate it with directed mutagenesis. While KNX-002 and Blebb binding involves the same structural elements found in this inner pocket: i.e. Switch-1 and Switch-2, HP-helix, HW-helix and the φLL-linker, the conformation of these drug binding sites strongly differs (Supplementary Movie 3). This is due to the fact that the two compounds target different structural states, and because of sequence polymorphism between class 2 myosins and PfMyoA. Unlike Myo2 inhibitors such as Blebb and its derivatives that recognize this inner pocket in the conformation it adopts in the pre-power stroke state,

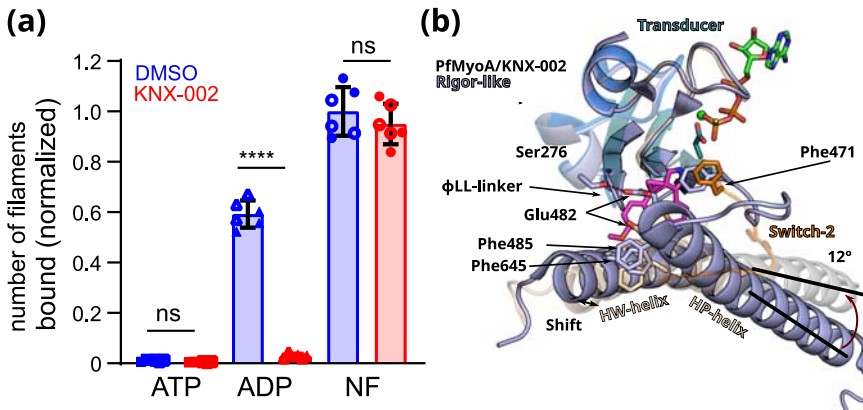

**Fig. 5 | PfMyoA.ADP binds actin weakly in the presence of KNX-002. a** KNX-002 weakens the affinity of actin for M.ADP but has no effect on binding in the presence of ATP or with nucleotide-free (NF)-PfMyoA. Number of actin filaments bound ± SD to surface immobilized PfMyoA (see Methods). See Supplementary Fig. 8 for examples of the raw data and Supplementary Fig. 9 for data with an expanded y-axis scale. The difference between ADP ± KNX-002 is significant ($p < 0.0001$) (one-sided ANOVA followed by Tukey's post-hoc test), but there were no significant differences between ATP ± KNX-002 ($p > 0.999$), nor between NF ± KNX-002 ($p = 0.64$). Data represent two experiments each performed in triplicate with independent protein preparations (open and filled circles). Data bars are mean ± SD. Source data are provided as a Source Data file. **b** The KNX-002 binding pocket does not exist in the Rigor state. PfMyoA/ATPγS/KNX-002 (colored by subdomains as in Fig. 2b) and PfMyoA in the Rigor-like state (light blue) (PDB code 6I7D[9]) are superimposed on the U50 subdomain and show how a change in the conformation and the orientation of the L50 subdomain would close the inner cleft and would thus not be compatible with KNX-002 binding. Indeed, the [L50]HP-helix rotates by 12° and the [L50]HW-helix shifts, changing the position of F485 and F645, two residues involved in KNX-002 binding. The reorientation of Switch-2 also leads to a new position for the key F471 residue that is incompatible with KNX-002 docking into this site in the Rigor state.

KNX-002 binds when myosin adopts the post-rigor conformation, which structurally differs greatly from the pre-powerstroke state (Supplementary Movie 3). KNX-002 thus inhibits PfMyoA using a previously undescribed binding mode requiring residues of the inner pocket in the post-rigor state. Based on sequence alignments (Supplementary Table 4), this binding pocket appears to be conserved in MyoA from other apicomplexan parasites such as *Toxoplasma* (toxoplasmosis), *Eimeria* (coccidiosis), and *Babesia* (babesiosis) and thus derivatives of KNX-002 could ultimately be applicable to treatment of a number of parasitic diseases. Consistent with this idea, KNX-002 was recently reported to inhibit the motility and growth of the *Toxoplasma gondii* parasite in culture[23].

Interestingly, transient kinetics and structural data both demonstrate that the mechanism of action of KNX-002 and Blebb greatly differ (Supplementary Table 3). While Blebb inhibition comes from the occupation of the inner cleft that efficiently blocks the initiation of the powerstroke, it does not interfere with the recovery stroke or ATP hydrolysis[20,28,32]. In contrast, KNX-002 steric occupation of the inner cleft in the post-rigor state pocket decreases ATP hydrolysis. KNX-002 indeed stabilizes a state of low affinity for actin in which binding to the inner cleft prevents the PfMyoA recovery stroke and prevents lever arm priming (Fig. 6). Our structure suggests that the molecular basis for reduced ATP hydrolysis includes slowing the recovery stroke and the inability of the motor to adopt the pre-powerstroke state, which is the most favorable to promote ATP hydrolysis by positioning a water molecule for in line attack of the γ-phosphate. Blebb and KNX-002 therefore belong to different categories of inhibitors with different mechanisms of action. KNX-002 represents a class of myosin inhibitors whose mode of action results from sequestering a post-rigor state. Trapping PfMyoA in a post-rigor state is sufficient to inhibit the activity since ATP hydrolysis does not occur in this state[33]. This result opens the door for the potential discovery of distinct inhibitors for different myosin targets as we report here an efficient way of preventing myosin from participating in force production.

The ability of KNX-002 to inhibit asexual blood stage growth in vitro demonstrates its antimalarial activity, and validates that PfMyoA and other essential myosins are potential antiparasitic pharmacological targets[9,10,24,31]. Moreover, KNX-002 was shown to not be toxic to human HepG2 liver cells at 80 µM[23]. While the IC50 for KNX-002 is in the micromolar range, these current data set a foundation for substantial further development with the structure of the PfMyoA bound to KNX-002 able to direct SAR that promote higher potency while conserving specificity. Thus, the structural data we provide paves the way to optimize this molecule for therapeutical potential. The biggest advantage of a strategy targeting PfMyoA with a KNX-002 derivative is that by inhibiting actin binding, it should efficiently slow motility and invasion phases of the parasite. A more potent derivative based on the KNX-002 scaffold (KNX-115) was recently characterized in vitro and in the *Plasmodium* parasite by James Spudich (CEO of Kainomyx) and colleagues[34]. KNX-115 shows great promise as a therapeutic agent because it is parasiticidal at multiple stages of the *Plasmodium* lifecycle, acts on resistant *Plasmodium* strains and displays no liver cell toxicity[34]. Structural studies of more potent KNX-002 derivatives will provide the blueprint for efficient PfMyoA inhibition and future development will aim to bring alternative prophylactic and/or symptomatic treatments. In addition, it will be essential to study how such potent inhibitors could bring a significant advantage either in conjunction with other antimalarials[35,36] or in conjunction with vaccination strategies that are unfortunately poorly efficient at present[6].

## Methods

### Key resources
Key resources used in this study are tabulated in Supplementary Table 5.

### KNX-002
KNX-002 (formerly CK2140597) was originally identified from high throughput screening of a library of 50,000 compounds at 40 µM final concentration (2% DMSO, v/v), performed by JS and EW at Cytokinetics, to identify inhibitors of the actin-activated ATPase of *T. gondii* MyoA (protein provided by Gary Ward, University of Vermont), and PfMyoA (protein provided by KMT). A coupled enzymatic ATPase assay[9] was used for screening. Actin was polymerized using 1 mM ATP, 2 mM MgCl$_2$, 20 mM KCl and left on ice for 30 min. Solution A was made with PM12 (12 mM Pipes-KOH, pH 7.0, 2 mM MgCl$_2$), 1 mM DTT, 0.1 mg/ml BSA, 0.4 mM pyruvate kinase/lactate dehydrogenase, 1 mg/ml Myosin XIV, 0.009% Antifoam (Sigma-Aldrich). Solution B was made with PM12, 1 mM DTT, 0.1 mg/ml BSA, 1 mM ATP, 1 mM

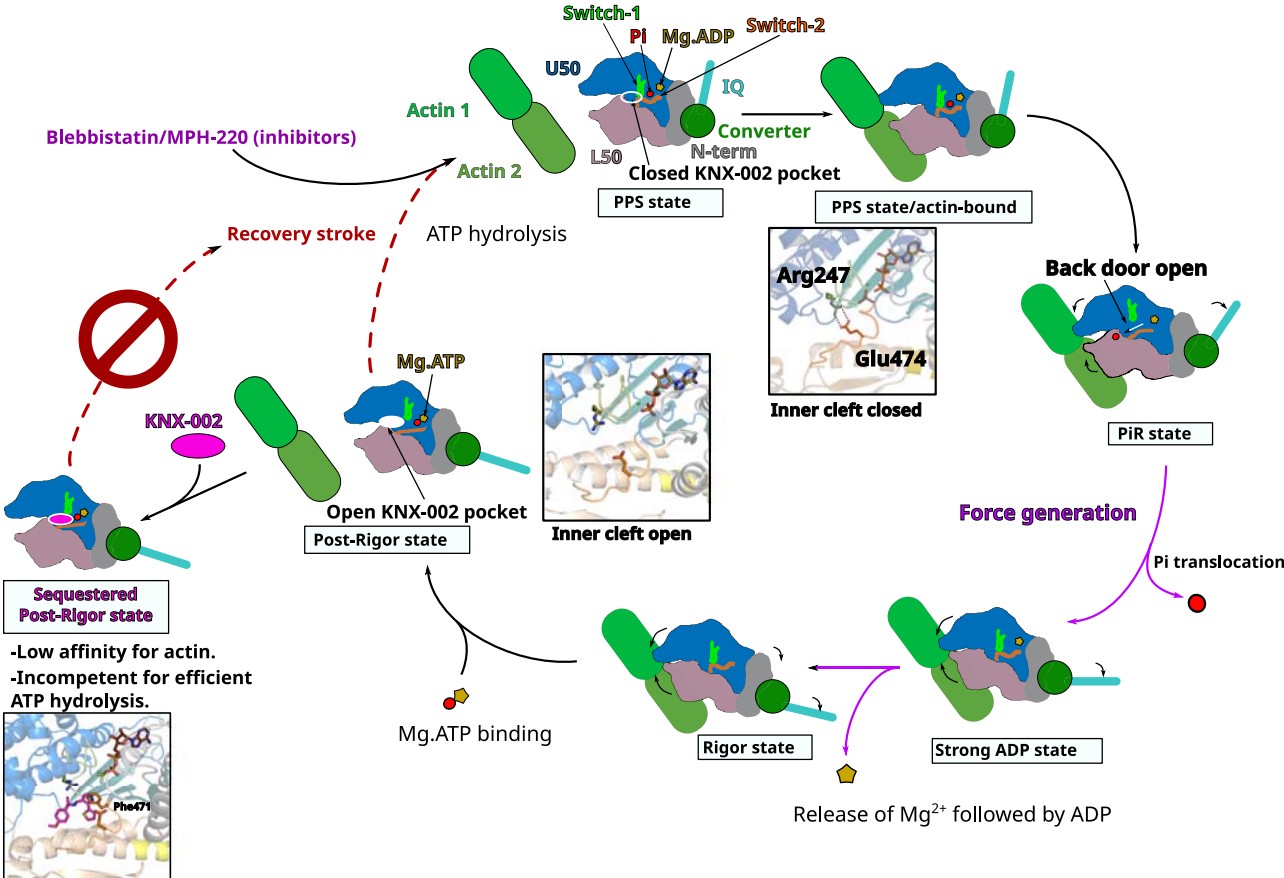

**Fig. 6 | Mechanism of the inhibition by compound KNX-002.** Schematic representation of the motor cycle of PfMyoA, where the state of nucleotide bound governs (i) the conformation of the motor and (ii) the affinity for the actin filament. In the post-rigor (PR) state, PfMyoA binds Mg.ATP and has low affinity for actin since the actin-binding cleft between U50 and L50 subdomains is open. After the recovery stroke that primes the lever arm, PfMyoA adopts the pre-powerstroke (PPS) state in which ATP is hydrolyzed. The weak association of the PPS to actin initiates a transition towards the $P_i$ release state ($P_i$R), the first force-producing state. This initiation of the powerstroke allows the opening of the $P_i$ release tunnel allowing $P_i$ translocation. After $P_i$ release, a large swing of the lever arm (powerstroke) leads to a Strong ADP state in which the actin-binding cleft is closed, allowing stronger association with actin. $Mg^{2+}$ and ADP are finally released from the Rigor state after a small swing of the lever arm and reorientation of the N-terminal subdomain. An ATP molecule can then bind, which leads to a fast transition towards the post-rigor state that detaches from actin. In contrast to the previously described Myo2 inhibitors, Blebbistatin and MPH-220[56] which target PPS, KNX-002 targets PR, which prevents the recovery stroke and ATP hydrolysis. When KNX-002 intercalates between the U50 and the L50, it stabilizes a state of poor affinity for F-actin that is not compatible with efficient ATP hydrolysis.

NADH, 1.5 mM phosphoenolpyruvate, 0.6 mg/ml actin previously polymerized and 0.009% Antifoam. 12.5 μl each of Solution A and B were added to a plate with a Packard MiniTrak liquid handler and then mixed at 2400 rpm for 45 s with a BioShake 3000 ELM orbital plate mixer (Qinstruments). The decrease in optical density at 340 nm as a function of time was measured on an Envision 2104 spectrophotometer (Perkin Elmer) with a 600 s data monitoring window sampled every 60 s. Hit progression included confirmation of activity in an independent experiment, inactivity against an unrelated ATPase (hexokinase), and dose-responsive inhibition. Hexokinase reactions were performed using glucose (1 mM) and sufficient hexokinase to generate ADP from ATP at a rate similar to the target myosins. Hit progression included confirmation of activity in an independent experiment, inactivity against an unrelated ATPase, and dose-responsive inhibition. Of note, KNX-002 was initially a weak hit in both campaigns due to a technical issue, but pursued for TgMyoA and later confirmed to be robustly active in both TgMyoA and PfMyoA assays, both with the initial library batch as well as with resupplied compound (>97% pure by LC/MS, Asinex). Kainomyx (www.kainomyx.com) licensed 6 compounds from Cytokinetics for drug discovery for malaria and other parasitic diseases, and James Spudich, CEO of Kainomyx, released CK2140597 (now KNX-002) without restrictions for academic studies in both the laboratories of Gary Ward, University of Vermont (for *Toxoplasma*), and KMT, University of Vermont (for *Plasmodium*). The KNX-002 used in this manuscript was custom-synthesized by Asinex (Winston-Salem, NC), and showed a high level of purity (97% pure), based on peak areas from a UV HPLC chromatogram at 275 nm using a Diode Array Detector.

## Protein expression and purification

The full length PfMyoA heavy chain WT and triple mutant (Phe270Tyr/Phe471Ala/Phe645His) were co-expressed with a UCS (UNC-45/CRO1/She4p) family myosin chaperone from *Plasmodium* spp. and two light chains PfELC and PfMTIP[37]. The full length PfMyoA heavy chain WT construct consists of the full *Plasmodium* myosin A heavy chain (PlasmoDB ID PF3D7_1342600/GenBank™ accession number XM_001350111.1) with a 13-amino acid spacing linker (NVSPATVQ-PAFGS) followed by an 88-amino acid segment of the *E. coli* biotin carboxyl carrier protein[38,39] and a C-terminal FLAG tag. The PfMyoA heavy chain triple mutant (Phe270Tyr/Phe471Ala/Phe645His) was created by site-directed mutagenesis using this WT heavy chain gene as a template. The baculovirus transfer vector pAcSG2 (554769, BD Biosciences) was used to incorporate the PCR product to make recombinant baculovirus. The *Plasmodium* UCS family myosin

chaperone is a chimeric clone that combines the coding sequence of *P. falciparum* (PF3D7_1420200/GenBank™ accession number XM_001348333.1) with regions from *P. knowlesi* (PKNH_1337800/GenBank™ accession number XM_002260772) in places where *P. falciparum* did not show consensus with other *Plasmodium* species. A Myc tag was added to the C-terminus and the resulting chimeric UCS chaperone PCR product was cloned into the baculovirus transfer vector pAcSG2 (554769, BD Biosciences) for recombinant virus production. The *PfELC* gene (PF3D7_1017500/GenBank™ accession number XM_001347419.1) was PCR-amplified from gBlocks® gene fragments (Twist Bioscience) and inserted into the pFastBac vector (10359016, Thermo Fisher Scientific) for recombinant virus production. The PfMTIP gene (PF3D7_1246400/GenBank™ accession number XM_001350813.1) was obtained from gBlocks® gene fragments (Twist Bioscience) and an N-terminal His$_6$ tag sequence added when cloning into the pAcSG2 vector (554769, BD Biosciences). For ΔN-MTIP, the N-terminal 60 amino acids were removed by site-directed mutagenesis, while retaining the N-terminal His$_6$ tag.

*Sf*9 cells (71104, Novagen) were infected with recombinant baculovirus and grown for 72 h in a medium containing 0.2 mg/ml biotin, harvested and lysed by sonication in 10 mM imidazole pH 7.4, 0.2 M NaCl, 1 mM EGTA, 5 mM MgCl$_2$, 7% (w/v) sucrose, 2 mM DTT, 0.5 mM 4-(2-aminoethyl) benzene-sulfonyl fluoride, 5 µg/ml leupeptin, 2 mM MgATP. An additional 2 mM MgATP was added prior to a clarifying spin at 200,000 × *g* for 40 min. The supernatant was then applied to an anti-FLAG-M2 affinity gel chromatography column (A2220, Sigma-Aldrich). The column was washed with 10 mM imidazole pH 7.4, 0.2 M NaCl, and 1 mM EGTA and the myosin eluted from the column using the same buffer plus 0.1 mg/ml FLAG peptide (A6002, APExBIO). The fractions containing myosin were pooled and concentrated using an Amicon centrifugal filter device (901024, Millipore), and dialyzed overnight against 10 mM imidazole, pH 7.4, 0.2 M NaCl, 1 mM EGTA, 55% (v/v) glycerol, 1 mM DTT, and 1 µg/ml leupeptin and stored at −20 °C.

Skeletal muscle actin was purified from chicken skeletal muscle tissue (frozen chicken breasts, Trader Joe's) by processing the muscle tissue into an acetone powder followed by purification of actin from the acetone powder[40]. Chicken breast tissue was ground with a meat grinder and suspended in 150 mM KPO$_4$ pH 6.7, 300 mM KCl, 2 mM EDTA, 1 mM DTT and 3 mM NaN$_3$. This slurry was added to a cheesecloth covered beaker and rinsed with 10 mM sodium bicarbonate and 0.1 mM CaCl$_2$. The tissue was washed with water and then with acetone and air dried overnight. The acetone powder was resuspended with G-buffer (5 mM Tris, pH 8.2 at 4 °C, 0.2 mM CaCl$_2$, 0.2 mM Na$_2$ATP, 0.5 mM DTT, 1 µg/ml leupeptin) and G-actin was extracted for 1 h at 4 °C, followed by centrifugation at 35,000 × *g* for 20 min. G-actin was polymerized to F-actin by addition of 2 mM MgCl$_2$ and 80 mM KCl (overnight, 4 °C). To remove tropomyosin, the KCl was increased to 600 mM, followed by centrifugation at 200,000 × *g* for 3 h. The pellet was resuspended in G-buffer and dialyzed against G-buffer extensively before a final clarification at 400,000 × *g* for 40 min. 2 mM MgCl$_2$ and 10 mM KCl were added to the supernatant to polymerize the G-actin. Actin concentration was determined by measuring the absorbance at 290 nm using an extinction coefficient of 26,600 cm$^{-1}$M$^{-1}$.

Skeletal myosin from chicken breasts (Shadow Cross Farm, Colchester VT) and cardiac myosin from rabbit hearts (41321, Pel-Freez Biologicals) were purified by extraction of ground tissue in 150 mM NaPO$_4$ pH 6.5, 300 mM NaCl, 5 mM MgCl$_2$, 2 mM MgATP, 2 mM DTT, 1 mM EGTA, 0.5 mM AEBSF, 0.5 mM TLCK and 5 µg/ml leupeptin[41]. Myosin was precipitated by a 20-fold dilution with water, and the precipitate resuspended and dialyzed in 25 mM NaPO$_4$ pH 7.0, 600 mM NaCl, 2 mM MgCl$_2$, 1 mM DTT, 1 mM NaN$_3$. 2 mM MgATP was added to the dialyzed myosin and the solution was clarified by centrifugation at 200,000 × *g* for 3 h. The supernatant was dialyzed against 0.2 M sodium pyrophosphate pH 7.0, 0.5 mM DTT and clarified at 40,000 × *g* for 20 min. The clarified supernatant was applied to a

DEAE-Sephacel ion-exchange column (17-0500-01, Amersham Biosciences) and eluted with a gradient of 0–0.5 M NaCl in 0.2 M sodium pyrophosphate pH 7.0, 0.5 mM DTT. Pure myosin fractions, identified by SDS-PAGE analysis, were dialyzed against 25 mM NaPO$_4$ pH 7.0, 600 mM NaCl, 2 mM MgCl$_2$, 1 mM DTT and 1 mM NaN$_3$. Glycerol was added to 50% and the myosin stored at −20 °C.

## Basal and actin-activated ATPase assays
ATPase activity was determined using a linked assay, which couples the regeneration of hydrolyzed ATP to the oxidation of NADH to NAD[+9]. 2 mM MgATP was added to a 300 µl aliquot of myosin (basal) or actomyosin (actin-activated), manually transferred to a cuvette, and the decrease in optical density at 340 nm measured as function of time on a Lambda 25 UV/VIS spectrophotometer (Perkin Elmer) with a 360 s data monitoring window sampled every 5 s. Data were fit with a four-parameter logistic curve (Graph Pad Prism v9.3.1). Conditions: 10 mM imidazole pH 7.5, 5 mM KCl, 1 mM MgCl$_2$, 1 mM EGTA, 1 mM DTT, 1 mM NaN$_3$, 2 mM MgATP, 5 nM PfMyoA, 55 nM skeletal myosin II, 100 nM cardiac myosin, 50 µM skeletal actin, 1% DMSO, 30 °C. Under these conditions, the actin-activated values for PfMyoA in the absence of inhibitor were 84.2 ± 1.4 s$^{-1}$ (*n* = 3) and 9.6 ± 3.4 s$^{-1}$ in the presence of 100 µM KNX-002.

## Motility and actin binding assays
Prior to in vitro motility, PfMyoA was spun at 350,000 × *g* spin for 20 min in the presence of 1.5 mM MgATP and a three-fold molar excess of skeletal actin. Solutions were added to a nitrocellulose-coated flow cell in 15 µl volumes in the following order. 0.5 mg/ml biotinylated bovine serum albumin (BSA) in buffer A (25 mM imidazole pH 7.5, 150 mM KCl, 4 mM MgCl$_2$, 1 mM EGTA, 10 mM DTT) was incubated for 1 min. 0.7 mg/ml BSA in buffer A was then incubated for 2 min. Neutravidin (25 µg/ml, Thermo Fisher Scientific) in buffer A was added for 1 min and rinsed out with three additions of buffer A. PfMyoA (120 µg/ml) was added in buffer A with either 1% DMSO or 100 µM KNX-002 and incubated 1 min before 3 washes with buffer A (with 1% DMSO or 100 µM KNX-002). Rhodamine-phalloidin labeled skeletal muscle actin was applied for 30 s followed by one rinse with buffer A (with 1% DMSO or 100 µM KNX-002) and one rinse with buffer B (buffer A plus 0.15% (w/v) methylcellulose, 25 µg/ml PfELC, 25 µg/ml PfMTIP, 50 µg/ml catalase (Sigma), 125 µg/ml glucose oxidase (Sigma), 3 mg/ml glucose). Buffer B containing 2 mM MgATP was added twice to initiate motility. From myosin addition onward, either 1% DMSO or 100 µM KNX-002 in 1% DMSO was present in all solutions. Actin filaments were visualized using an inverted microscope Zeiss Axiovert 10), a Rolera MGi Plus digital camera and the Nikon NIS Elements software package. Filament speeds were tracked and analyzed with the Fast Automated Spud Trekker analysis program (FAST, Spudich laboratory, Stanford University). This online software is free and available for download at http://spudlab.stanford.edu/fast-for-automatic-motility-measurements. Speeds were fit to a Gaussian curve.

Actin binding to myosin was measured with a modified in vitro motility assay. For visualization in apo, ADP or ATP buffers, motility buffer B that contained either no nucleotide, 2 mM MgADP or 2 mM MgATP without methylcellulose was added after the rhodamine-phalloidin actin binding step. The number of actin filaments bound was quantified manually. Conditions: 25 mM imidazole pH 7.5, 150 mM KCl, 4 mM MgCl$_2$, 1 mM EGTA, 10 mM DTT, 0% methylcellulose, 1% DMSO, 30 °C. 2 mM MgATP or MgADP were added for +ATP or +ADP condition, respectively.

## Phosphate burst
Chemical hydrolysis of ATP by myosin was performed by manual mixing under single turnover conditions. PfMyoA (25 µM) was pre incubated with 1% DMSO or KNX-002 (100 µM in 1% DMSO), manually mixed with 20 µM MgATP, aged for 5 s, and then quenched by addition

of 0.3 M perchloric acid before quantifying free phosphate as follows[42]. To a 50 μl sample, 200 μl Detection Solution (5.72% (w/v) ammonium molybdate in 6 N HCl, 2.32% (w/v) polyvinyl alcohol and 0.0812% (w/v) malachite green) was added and allowed to incubate for 1 min before adding 50 μl 34% sodium citrate. A phosphate standard curve (0-2 nmoles) was used to convert the optical density signal change at 595 nm to nmoles phosphate. Controls using myosin, dialysis buffer or ATP alone showed undetectable phosphate. Conditions: 10 mM imidazole pH 7.5, 150 mM KCl, 4 mM $MgCl_2$, 1 mM EGTA, 10 mM DTT, 25 μM PfMyoA, 20 μM MgATP, 1% DMSO, 30 °C.

### Actin polymerization assay
Monomeric PfAct1 was polymerized in a buffer containing 10 mM imidazole, pH 7.5, 50 mM KCl, 2 mM $MgCl_2$, 1 mM EGTA, 0.25% methylcellulose, 2.5 mM MgATP, 10 mM DTT, 0.13 mg/mL glucose oxidase, 50 μg/mL catalase, 3 mg/mL glucose and either 1% DMSO or 100 μm KNX-002 at 37 °C. Filament growth was visualized using 0.5 μM actin–chromobody Emerald (ChromoTek) and quantified as a function of time[43]. TIRF microscopy recorded with a Nikon ECLIPSE Ti microscope and an Andor EMCCD camera (Andor Technology) run by the Nikon NIS Elements software package. Samples were excited with a 488-nm TIRF-field laser line and emission was observed with a 525/50 filter.

### Transient kinetics
All stopped-flow experiments were performed on a KinTek stopped-flow apparatus (KinTek Corporation, model SF-2001). All concentrations stated are after mixing into the stopped-flow cell, except where stated. Light scattering at 295 nm with a 295 nm interference filter orthogonal to the incident light was used to monitor the dissociation of actomyosin (0.2 μM actin and 0.15 μM PfMyoA). Mant-ATP and mant-ADP binding to myosin was measured by exciting at 290 nm and monitoring emission light that passed through a 400 nm long pass filter, so that only nucleotide bound at the active site is monitored via energy transfer from a Trp. For mant-nucleotide binding experiments in the presence of inhibitor, 100 μM KNX-002 and 1% DMSO were present in both syringes. All traces were analyzed using the software package provided by KinTek and fit to either single or double exponentials. Multiple time courses (usually 3-8) were averaged prior to curve fitting. Conditions: 10 mM HEPES pH 7.5, 50 mM KCl, 4 mM $MgCl_2$, 1 mM EGTA, 1 mM DTT, 1% DMSO, 30 °C.

### Crystallization, data processing, structure determination and refinement
Full-length PfMyoA with bound light chains PfELC and MTIP-ΔN (lacking residues 1-E60) was co-crystallized with or without KNX-002. PfMyoA/ELC/ MTIP-ΔN at 10.6 mg/ml was incubated with 2 mM Mg.ADP-gamma-S (ATPγS) for 20 min (Apo-PfMyoA). PfMyoA-KNX-002 crystals were obtained with and without the incubation of 2 mM Mg.ADP-gamma-S (ATPγS) for 20 min and 0.5 mM KNX-002 for 40 min. All the samples were centrifuged at 11,000 × g for 15 min at 4 °C before crystallizing assays. Both crystals of PfMyoA/ELC/MTIP-ΔN were obtained at 4 °C by the hanging drop vapor diffusion method from a 1:1 (v:v) of protein and mother liquor. The crystallization buffer is 2.0 M ammonium sulfate, 0.1 M sodium HEPES pH 7,5, 4% PEG400. The crystals were transferred in the cryoprotection containing the mother liquor +25% glycerol (v:v) and immediately flash frozen in liquid nitrogen.

X-ray datasets were collected at the ESRF Synchrotron, on the ID30B beamline[44]), at 100 K with the following parameters: PfMyoA/ATPγS/Apo (λ = 0.9677 Å, 46.8% transmission, flux start 7.54e[11] ph/sec); PfMyoA/ATPγS/KNX-002 (λ = 0.9687 Å, 1.2% transmission, flux start 9.89e[12] ph/sec); PfMyoA/NF/KNX-002 (λ = 1.07216 Å, 30.6% transmission, flux start 6.83e[11] ph/sec). Diffraction datasets were processed using XDS[45] and AutoPROC[46]. Both crystal forms (PfMyoA/ATPγS/Apo), (PfMyoA/ATPγS/KNX-002) and (PfMyoA/NF/KNX-002) belong

to the space group $P2_12_12_1$. The molecular replacement solution was obtained by using Phaser[47] using the PDB coordinates from 6YCY[10] without solvent and water as model target. The 3D structure of compound KNX-002 and the related restraints were generated using Elbow[48]. The models were manually refined using Coot[49]. The refinement was performed using Buster[50] and Phenix[51]. The final Ramachandran for favored, allowed and outliers are 97.25%, 2.65%, 0.09% for (PfMyoA/ATPγS/Apo); 96.51%; 3.49%, 0% for (PfMyoA/ATPγS/KNX-002) and 97.15%; 2.85%, 0% for (PfMyoA/NF/KNX-002), respectively. The atomic coordinates and structure factors have been deposited in the Protein Data Bank[52], www.pdb.org, with accession numbers 8A12 (PfMyoA/ATPγS/Apo), 8CDQ (PfMyoA/ATPγS/KNX-002) and 8CDM (PfMyoA/NF/KNX-002).

### Analysis of the drug binding pockets
The residues involved in drug binding were automatically determined with the software LigPlot+[53]. Default set up interaction cut off at 3.9 Å, but we investigated manually longer range interactions (yet <5 Å) by visualization with the PyMOL software[54]. The figure panels were created and manually edited with LigPlot+.

### Effects of KNX-002 on parasite asexual blood-stage growth
Asexual parasite blood stage growth (measuring the ability of parasites to invade, replicate, exit and reinvade erythrocytes) was undertaken using the SYBR Green asexual growth assay, undertaken largely as published[55]. In brief 96-well black clear bottom plates (Corning) were pre-printed with test compound and normalized with DMSO (Merck) to 0.5% of a total assay volume of 100 μl. Highly synchronized ring stage parasites and blood was added to each well so that the final parasitaemia was 2% and the hematocrit was 1%. The compound was incubated with parasites for 72 h before being frozen at −20 °C (to aid with cell lysis). The plate was thawed on ice and a lysis buffer (20 mM Tris pH7.5 (Merck), 5 mM EDTA (Merck), 0.008% w/vol saponin (Merck), 0.08% vol/vol triton-x100 (Merck) and SYBR-Green I (Thermo Fisher Scientific) at a final concentration of 0.02% vol/vol). The 96-well plate was incubated for 1 h at RTP before each well was assayed for fluorescence using GFP filters (Excitation 485 nm/Emission 535 nm) on a microplate reader (TECAN).

### Reporting summary
Further information on research design is available in the Nature Portfolio Reporting Summary linked to this article.

## Data availability
The atomic models and crystallographic data of PfMyoA/ATPγS/Apo, PfMyoA/ATPγS/KNX-002 and PfMyoA/NF/KNX-002 have been deposited on the PDB under the accession codes 8A12, 8CDQ and 8CDM, respectively. Reference structures used in this work are 1YV3, 6I7D and 6YCY. Source data are provided as a Source Data file. Source data are provided with this paper.

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

## Acknowledgements

This work was supported by NIH grant R01AI 132378 to K.M.T. and A.H., a grant from the Human Frontier Science Program (RGY006/2016) to J.B. and A.H., grants from the Wellcome (Investigator Award to J.B., 100993/Z/13/Z) and the Medicines for Malaria Venture (RD-08-2800) to J.B. We thank Cytokinetics for providing purified actin for the screen and access to their high-throughput screening library, infrastructure and staff. We are grateful to Dr. James Spudich (CEO of Kainomyx) for giving us the opportunity to study KNX-002 in the *Plasmodium* system via a Continuing Research Agreement between Kainomyx and the University of Vermont (K.M.T.), and Kainomyx and the Institut Curie (A.H.). We acknowledge the ESRF for providing in-house research beamtime on the ID30B, especially Dr. Andrew McCarthy. We thank Dr. James Spudich and Dr. Kathleen Ruppel for critical reading of the manuscript, and Dr. James Spudich for useful suggestions regarding the motility experiments. We thank Dr. Gordon Leonard (ESRF, Grenoble, France) for critical reading of the manuscript, and Dr. Hailong Lu (University of Vermont) for performing the TIRF polymerization assay on PfAct1.

## Author contributions

A.H. and K.M.T. designed and directed the research. D.M. crystallized PfMyoA and PfMyoA complexed to KNX-002. D.M. and C.M.D. collected the data. D.M. and J.R.P. processed the data, solved and refined the structures. J.P.R. performed the in vitro functional assays. P.M.F. and J.E.M. expressed and purified PfMyoA. F.F. performed the parasitemia assays. J.S. and E.W. developed the screening assay and performed the high-throughput screen and hit characterization for CK2140597/KNX-002 while at Cytokinetics. D.M., J.P.R., D.A., J.R.P., K.M.T. and A.H. analyzed and discussed the results. D.M., J.R.P. and A.H. wrote the initial version of the paper, with the help of J.P.R., D.A., J.B. and K.M.T. All the authors were involved in reviewing and editing the paper. J.B., J.R.P., K.M.T. and A.H. were involved in project administration. J.B., K.M.T. and A.H. provided funding.

## Competing interests

A.H. receives research funding from Cytokinetics and consults for Kainomyx. All other authors have no competing interests.
