## [Peer Review File · Nature Communications]

Mechanism of small molecule inhibition of Plasmodium falciparum myosin A informs antimalarial drug designREVIEWER COMMENTS

Reviewer #1 (Remarks to the Author):

The paper by Moussaoui et al. reports the discovery and characterization of a novel *Plasmodium falciparum* myosin A inhibitor that the authors hope will serve as a template for future antimalarial drugs. The paper describes a series of biochemical and structural studies that reveal a novel mechanism of motor inhibition and map the binding site of this new inhibitor. The experiments are thoughtfully designed, and the paper is well written. Although drug development is at an early stage, the discovery of a novel site to target is impactful and will be of interest to the field.

1- Can Knx002 bind its active site when MgATP (or MgADP) is already bound (as shown in Fig. 6), or does inhibitor binding have to precede nucleotide binding? I think this is an important question, since if binding needs to precede nucleotide binding, it will need to bind a myosin in the near-rigor conformation in the absence of nucleotide while detached from actin. This would be a rarely populated state. Please comment in the text either way.

2- The experiments in Supp Fig. 1 measure the affinity of Knx002 to apo and MgADP-myosin A by quantifying the fraction of a slow kinetic phase of the MantATP binding transient. The finding that there are two resolvable kinetic phases suggests that the off rate of Knx002 is very slow. This is a point worth mentioning in the text. Also, is it possible that the affinities measured in Supp Fig. 1 panels (a) and (b) are both measuring the affinities of Knx002 for apo myosin, i.e., the MgADP dynamically associates, so Knx002 can bind the apo state. If the inhibitor binding precedes nucleotide binding, then one might expect the affinities in the two experiments to be nearly the same.

3- The rate of the slow phase in Supp Fig. 1 (a) is reported to be 1/s. According to the rate constant reported in Fig. 4, it should be 1.6/s. There are no uncertainties in Fig. 4, so I don't know if this is within the experimental error. Alternatively, it is possible that the curve in Fig. 4 "plateaus" at higher concentrations. Please comment.

4- I greatly appreciate the details provided in Supp Figs. 3 and 4 and in Supp Table 1. They are useful for understanding the structures and their relationships to other myosin isoforms. However, I think it would be appropriate for the authors to compare the residues required for Knx002 binding to other members of the myosin superfamily. It is not surprising that the myosin A Knx002 binding pocket is very different from class-2 myosins. It would be of additional interest and importance to compare the sequences to unconventional myosins that show more homology to myosin A. I did a brief survey of sequences, and myosin A does seem to be divergent at these sites – so it would give the reader confidence in the selectivity of this inhibitor.

5- I was not able to get the Supp movies to open on my Mac. I did not try a Windows computer.

Reviewer #2 (Remarks to the Author):

This is an important piece of work, describing for the first time a lead compound specifically targeting *Plasmodium falciparum* myosin A (MyoA), which is the key component and motor of the molecular complex responsible for generating force for the malaria parasite gliding motility and host cell invasion. The manuscript describes a new compound, Knx002, that specifically inhibits the basal and actin-activated ATPase activity of MyoA, having no effect on cardiac or skeletal Myo2. Two crystal structures of MyoA at resolutions of 2.0 and 2.1 Å in the apo-ADP form and in complex with the inhibitor Knx002 and the poorly hydrolyzable ATP analog ATP-γ-S, respectively, are presented and analyzed. In addition, effects of Knx002 on the affinity and kinetics of MyoA binding to nucleotides and actin are reported. The work should be published and will have an impact on the field and also open possibilities for drug design against an important global health threat. However, the manuscript needs additional work and some statements/conclusions should be either better backed-up, toned down, or reconsidered before I could recommend its publication.

I will list my specific comments below, divided into major, minor or more technical, and typographic/linguistic/stylistic points.

Major points:

- The authors spend a lot of words describing the novelty of the Knx002 binding pocket in MyoA. The question is: Can the pocket really be considered "novel"? It seems from the figures and the residues involved in the binding site that it is very much the same binding site previously described for blebbistatin, albeit the binding mode is different. The statements on the "novel binding site" should be toned down or the authors should convince the reader how the pocket, not just the binding mode, is new.

- The mechanism of inhibition of ATP hydrolysis is stated to be the impact of the compound on the Mg²⁺ coordination. Although the data do suggest that as a possibility, it is not possible to come to such a conclusion without presenting a catalytic mechanism and having structures of MyoA in both ATP (or ATP- γ -S) and ADP form in the presence and absence of the compound. Comparing apo-ADP-MyoA and Knx002-ATP- γ -S-MyoA and ATP, ATP- γ -S, and ADP structures of another myosin (Myo2) and the simulations reported are not direct comparison that can be used to firmly draw this conclusion. Either the authors need to perform additional experiments to confirm their hypothesis and propose a detailed catalytic and inhibition mechanism or the conclusion has to be toned down. Because Knx002 inhibits hydrolysis, it should also be possible to get an ATP structure with the compound.

- In all the actin-related assays, vertebrate skeletal muscle actin has been used. Why not use parasite actin, as the authors are in a unique position by having access to recombinant *P. falciparum* actin, on which they have published high-impact articles (e.g. Robert-Paganin et al. Nat Comm 2021; Lu et al. PNAS 2019)?

- It seems that it is not possible to draw conclusions on the in vitro motility assays if almost no filaments could be observed in the presence of the compound. This is both stated in the manuscript and visible in the two supplementary movies. Given this, it is strange that the effect of the compound on actin polymerization is not discussed and not characterized. This is a critical point, given that the assays were performed using skeletal muscle actin (see above), and it seems that the lead compound actually may inhibit host actin polymerization. Could this also be a reason for the toxicity of the compound in cell culture experiments (see below)?

- The compound has a low affinity and also significant toxicity at the high concentrations used in the assays. In the text, the authors quote survival rates of three different cell types (Supplemental Table 1) at a Knx002 concentration of 20 μ M. At this concentration, the fibroblast and epithelial cells seem unaffected, but ~20% of the hepatic cells died. The assays have been performed at 100 or 200 μ M concentrations. At 100 μ M, only the epithelial cells remained unaffected, whereas ~20% of the fibroblasts and ~70% of the hepatic cells died. At 200 μ M, basically all cells died (except for ~20% of the fibroblasts). Why are the cell survival numbers given for much lower concentrations than used in the assays and why is this toxicity not considered an issue? 100 and 200 μ M concentrations are tens of times IC₅₀. Were lower concentrations tested? What happens at IC₅₀? Is there any measurable effect?

- The triple mutant has a large effect on the sensitivity of the mutant to the compound in the ATPase assay. Were the effects of these mutations on the folding and stability of the protein assessed in any way? It would be also very informative to see the active site configuration of this mutant protein.

- This may already be out of the scope of this work, but of course ideally, compounds like this should also be evaluated in a mouse model.

Minor or technical points:

- Why are the diffraction data so incomplete? Because of anisotropy? This can have an impact on the quality of the ED maps. Low completeness in the low resolution shells can cause distorted maps or missing parts in the electron density. Given this and without having access to the data or

the maps, it is hard to say whether it can with certainty be said that the apo structure has ADP and not disordered ATP-gamma-S. The completeness of the data sets (in Supplementary Table 2) is given as spherical completeness and with ellipsoidal correction. However, it is not clear if the rest of the statistics given are for the non-correct or corrected data? What correction method was used? Which data were used for refinement and map calculation? Did the ellipsoidal correction have an impact on map quality - could important features be seen in both maps?

- Why is the number of replicates only 2 in many of the experiments? Usually, triplicate series should be expected.

- What is n in Figure 1c?

- The cell survival assay methods are not described at all.

- The purification of cardiac and skeletal myosins are not described. The purification of skeletal muscle actin is described under the title "Myosin expression and purification". The title should be "Protein expression and purification", and those should be described for all the proteins used.

- The description of the phosphate burst assay is minimal. Either references to literature should be provided or the method described in sufficient detail, allowing the reader to understand exactly how it was done. How was the possibility of the very common PO₄ contamination from purification and reagents taken into account?

- In the transient kinetics assays, 3-8 traces were measured. A common strategy is to measure at least 10 traces, so that there still remains a large enough number of observations in case some need to be excluded as outliers. Would 3 traces mean that most of the data were excluded as outliers?

- Also, in the transient kinetics assay, the protein concentrations were fairly low. Possibly this is not a problem with the fluorescent nucleotide analogs, but why is the ratio of actin:myosin not 1:1? In Figure 4a, there are no error bars or standard deviations given.

- The simulations are poorly described. It seems, based on the methods description, that the Mg²⁺ ion was placed in the "expected" position from the ADP structure at the start of the simulations. At the timescale used (220 ns), it is probably not to be expected that it would move, and the side chains would be more likely to move to accommodate it. With today's computing power, simulations in the ms scale are not out of reach. Can the used force field handle divalent cations, which have been notoriously challenging for simulations?

- In Supplementary Figure 1, it seems that there are no error bars in panel (a). Are they missing or are the errors so small that they are invisible? This is one of the experiments, where only duplicates were performed/used.

- In the text, U50 and L50 should be explained to readers not so familiar with myosin structure.

- In the text: "Most compounds to date target the PPS state" - I suppose this refers to myosin inhibitors in general? It would be good to state this more clearly.

- The final concentration of DMSO in the assays is not always very clearly stated. Is it always 1% or below, as 1% was used in the control experiments?

Typographic, linguistic, stylistic, etc. issues:

- The authors refer to MyoA as "atypical". This might suggest that MyoA is not a typical myosin of its class (class XIV). Unconventional would seem like a better choice of wording.

- The sentence: "627,000 people died of malaria in 2020, the majority being children under the age of 5 years". Better would be not to start a sentence with a number.

- In the methods section "Myosin expression and purification": What are PUNC chaperones? Should

it be "PfUNC"? In any case, it should be spelled out.

- In the methods section "Myosin expression and purification: "constructs were purified"... "Construct" usually refers to the DNA construct encoding the protein. This should be reworded.
- Figure 1b is strange and not explained well in the figure legend.
- There is a dot missing after "Knx002" on line 4 of the legend of Figure 2.
- Figures 6 and S2 seem very similar and to a large part redundant. Figure S2 probably was included so that it can be referred to early in the text, while still having Figure 6 as the last one. It might be worth considering making these more different from each other.
- The colors in Figures 2, 3, and 5 could be improved. The colors, especially the blue and green/cyan shades, are difficult to tell apart. It would be better to use colors that differ more clearly from each other.
- Some of the color names/codes used in the figure legends are not real color names but rather codes used in the programs used to make the figures. It would be better to use e.g. "green" instead of "deep teal cyan", "blue" (or "dark blue") instead of "marine blue", "beige" instead of "wheat", etc.
- On line 11 of the legend of Figure 3, "conserved residues" should be in singular, as only one residue is in a purple box.
- On lines 15-16 of the legend of Figure 3, the word "binding" seems to be missing after "Blebb".
- On the 9th line of the legend of Figure 5, the word "(colours)" probably should be replaced by the actual colors used in the figure.
- The style of the figures in general could be improved and unified; Some of the panels are in boxes, some not, the order of the panels is not always logical, there's a mixed use of bold and normal fonts in the figures, some figures have text elements which are on the border of being too small to read.
- It seems that Supplementary Table 4 is not referred to in the text.
- The accuracy of the rmsd values for the superposition of the structures with three decimals seems a bit exaggerated. Also, the value given is not exactly the same in the main text and the figure legend.

Reviewer #3 (Remarks to the Author):

In this study, the Authors characterize in detail a mechanism of action of a previously identified inhibitor of myosin A from Plasmodium (PfMyoA). PfMyoA is part of glideosome that is critical for the parasite mobility and infectivity, and is a validated drug target. The Authors combine multiple experimental and computational methods to characterize the selected compound action from the level of protein structure to the parasite level. In my opinion, this is a very convincing, coherent study, which is additionally clearly written. The conclusions are consistent with the results. The results seem very useful for anti-Plasmodium structure-based drug design. I would recommend to publish this work after addressing the remarks below.

Major remarks:

the "Methods" section:

It would be good if the Authors provided more information about the system setup for MD simulations:

- I. 527-528: "Starting from the PfMyoA/Apo and PfMyoA/Knx002 coordinates, ATP was modelled

after ATPγS and the Mg²⁺ was positioned as found in PfMyoA/Apo." - Could you provide more details how Mg²⁺ was modelled in the liganded system? How well the surrounding of Mg²⁺ can be aligned in the two systems? Maybe the alignment could be shown in SI?

- How Mg²⁺ and ATP were parametrized for MD? (which parameters?)
- Was the hydration shell of Mg²⁺ from the X-ray preserved in the built MD systems?
- How large were the systems? (no. of atoms in total and waters)
- What was the simulation protocol?

Minor remarks:

- In my opinion, it would be good if the Authors introduced blebbistatin in the Introduction, since it is one of the main compounds analyzed (why is it used as a reference?).

- the sentence l. 135-138 is too long and unclear:

"Whether bound to Knx002 or not, PfMyoA crystallized in the post-rigor (PR) state, an ATP-bound myosin structural state with low affinity for the actin track which is populated upon detachment of the motor from the track prior to the priming of its lever arm (Supplementary Fig. 2)."

- l. 95-97: the sentences: "Because PfMyoA is essential for blood cell invasion 9,10, we tested the effect of Knx002 on P. falciparum asexual, blood-stage growth, itself dependent on the ability of merozoites to invade erythrocytes 23. Knx002 inhibited asexual blood stage growth of merozoites ..."

are unclear to me. Could the Authors rephrase or expand these sentences?

l. 174: "bonds" -> "interactions"?

l. 248: Mant-ATP - for some readers it may be obvious, but the Authors could mention what is Mant-ATP and why it is used.

l. 335: what is "A.M.ADP"? This abbreviation seems not to be introduced.

Figure 1:

(b) The dotted lines on the chemical structure are too thin to differentiate colors. The bottom part of the subfigure is not explained in the caption (I think the cycle should be briefly explained).

Figure 2:

There should be more space between the subfigures to make the figure more clear.

(c) The two structures should be shown in different colors, not colored by domain, because now conformational differences are hard to be seen or the subfigure should be skipped (RMSD would be enough). RMSD: it should be specified that it is Cα RMSD.

Figure 3:

(b) Are interaction identification criteria defined somewhere? Were the interaction diagrams generated by some software?

(d) Closing bracket is missing.

(f) "Residues involved in Blebb are circled" - the word "binding" is missing.

l. 375-377:

"Importantly, it is the first time that a compound is reported to bind to the inner pocket of a Post-rigor myosin state, without the requirement for much conformational change." - this sentence is a bit unclear: conformational change of what?

Overview of changes made to the manuscript

We found the reviewers' comments helpful and have edited the manuscript accordingly (specific responses below). Note that we have changed the inhibitor nomenclature throughout to KNX-002 per the request of Kainomyx.

Overall, most comments of the reviewers were readily addressed and show that they appreciate the novelty of the data and its significance. But questions raised by Reviewers 1 and 2 led us to collect additional kinetic and structural data. Based on this, we significantly changed one section of the manuscript formerly called "**KNX-002 slows ATP binding and affects Mg²⁺ coordination in the active site**" that is now entitled "**Characterization of the impact of KNX-002 on the nucleotide binding site**".

We obtained additional data to more thoroughly respond to questions regarding the ability of KNX-002 to prevent ATP hydrolysis, to investigate its ability to bind when nucleotides are bound and the reasons why KNX-002 affects nucleotide binding. Our additional data indicated that despite the fact that ATP analogs (ADP.BeFx and ATPγS) are usual in crystallography to trap states of the motor that correspond to ATP bound, the two analogs provided different observations regarding the ability of KNX-002 to influence Mg²⁺ coordination. The ATP analog first used in our study (unpublished dataset with ADP.BeFx) disrupted the hexa-coordination of Mg²⁺ ion that is typical of myosin active site, and influenced the refinement of our original ATPγS dataset. Using a better ATPγS dataset and removing model bias clearly showed that ATPγS does not disrupt the hexa-coordination of the Mg²⁺ ion. As ATPγS is a closer analog of ATP than BeFx is, our conclusion resulting from additional functional and structural data is that ATP favors hexa-coordination around the Mg²⁺ as is the case in the absence of drug. Crystallization with ADP.BeFx had led us to conclude that there was a perturbation of the hexa-coordination, but this is only true for this non-physiological nucleotide analog. We have thus revised the manuscript accordingly.

We have also provided new data in Figure 4a for the binding of mant-ATP and mant-ADP to PfMyoA in the absence or presence of KNX-002 over a broader range of nucleotide concentration at 4 mM MgCl₂ using newly purchased nucleotide stocks, which we believe to more reliably report the change in association rate constant induced by KNX-002 (~a 3-5-fold slowing) for both nucleotides.

In the revised manuscript, we can now present the following structures that are physiologically relevant or that provide answers to the reviewer's remarks.

- PfMyoA/KNX-002 co-crystallized with ATP-γS (with Mg²⁺ hexa-coordinated)
- PfMyoA/apo (no compound) co-crystallized with ATP-γS, which leads to ADP bound and Mg²⁺ hexa-coordinated.
- PfMyoA/KNX-002 without nucleotide bound (nucleotide-free, NF condition).

In the detailed response to reviewers that follows, we describe another dataset that was collected but that will not be presented in the manuscript because it does not add to the understanding of the mechanism of the compound, although it helps address some of the reviewers' concerns.

PfMyoA/KNX-002 soaked with ATP indicates how ATP would bind to the active site when KNX-002 is bound to the motor. It also shows that ATP is not hydrolyzed.

[Note that it is not possible to co-crystallize KNX-002 with ATP as the affinity of the drug does not allow it to stay in place to prevent hydrolysis for the long period of time (one week) required for the appearance of crystals.]

REVIEWER COMMENTS

Reviewer #1 (Remarks to the Author):

The paper by Moussaoui et al. reports the discovery and characterization of a novel Plasmodium falciparum myosin A inhibitor that the authors hope will serve as a template for future antimalarial drugs. The paper describes a series of biochemical and structural studies that reveal a novel mechanism of motor inhibition and map the binding site of this new inhibitor. The experiments are thoughtfully designed, and the paper is well written. Although drug development is at an early stage, the discovery of a novel site to target is impactful and will be of interest to the field.

We thank Reviewer 1 for the positive comments and helpful suggestions.

1-Can Knx002 bind its active site when MgATP (or MgADP) is already bound (as shown in Figure 6), or does inhibitor binding have to precede nucleotide binding? I think this is an important question, since if binding needs to precede nucleotide binding, it will need to bind a myosin in the near-rigor conformation in the absence of nucleotide while detached from actin. This would be a rarely populated state. Please comment in the text either way.

We agree with the reviewer that if KNX-002 would require binding to a nucleotide-free head prior to allowing ATP binding, then binding would occur in only a very small fraction of the heads when they are cycling. Indeed, cryo-EM data and other studies have shown that ATP binding to the rigor state precedes efficient detachment, and detachment leads to the post-rigor state. Our crystallization studies indicate that the conformational state the drug binds is post-rigor. Thus, most of the heads that populate the post-rigor state when myosin cycles on actin have ATP bound.

Both our functional and structural data indicate that KNX-002 binding does not need to precede nucleotide binding, and that KNX-002 can bind to a head that has bound nucleotide.

- 1) **Functional argument:** The observation that KNX-002 inhibits actin-activated ATPase activity and parasite blood stage growth indicates that the compound works to inhibit myosin even when physiological concentrations of nucleotide are present, which means that it is unlikely to be a state poorly represented in the population of heads. We added the following sentence at the end of paragraph 2 of Results (l. 120-122):

“The observation that KNX-002 inhibits actin-activated ATPase activity and parasite blood stage growth indicates that the compound inhibits myosin when physiological concentrations of nucleotide are present.”

We also performed an experiment where we mixed 200 μM KNX-002 with PfMyoA.mant-ADP and observed a small increase in bound mant-ADP fluorescence at a rate of $\sim 1 \text{ s}^{-1}$ ($n=2$), consistent with the ability of KNX-002 to bind to PfMyoA that has mant-ADP bound at the active site. The observed signal must come from bound nucleotide because we excite the mant-nucleotide via energy transfer from a Trp residue in PfMyoA. This new experiment is described in the last sentence of the legend to Supplementary Figure 1 (b) where the affinity of KNX-002 for M.ADP is measured. This observation supports the idea that KNX-002 can bind if ADP is already present and that we are thus measuring the affinity of KNX-002 for an ADP and not a nucleotide-free state.

- 2) **Structural data** also support the idea that KNX-002 can find its binding pocket whatever the nucleotide bound, because NF, MgADP and MgATP are all compatible with the post-rigor state:

we have been able to obtain data sets from heads with both KNX-002 bound and any of these active site occupation states. This shows that the changes required for KNX-002 binding are small and localized around the nucleotide binding site, not the ligand binding site (see **Supplementary Figure 6**).

We now provide an additional structure of PfMyoA in the nucleotide-free state bound to KNX-002 and show that the structure is similar to that found when ATP γ S is bound. These additional data along with functional data are described in a revision of the previously entitled section: “**KNX-002 slows ATP binding and affects Mg²⁺ coordination in the active site**”, which is now entitled “**Characterization of the impact of KNX-002 on the nucleotide binding site**”.

We removed the molecular dynamics and description of the consequence of a possible modification of the Mg²⁺ ion coordination as we now find it to be irrelevant for the mechanism of the compound in the presence of ATP. Both our original kinetic data and newer data indeed indicate that Mg²⁺ concentration (4 mM versus 50 μ M) has little effect on ATP binding to the active site both in the absence or presence of KNX-002. We also find that Mg²⁺ ATP coordination is not disrupted by KNX-002 when ATP γ S is bound in the active site, unlike what is seen with ADP.BeFx. It seems that the active site and drug binding sites do not require or prevent each other from binding their respective ligand. Consistently the access of the compound to the ligand binding site and the conformational changes required to bind KNX-002 are similar when no nucleotide is present or when ATP γ S is already bound. In the motor cycle, the heads that adopt the post-rigor state are bound to ATP, and thus it is likely that KNX-002 inhibits motor activity while ATP is already bound to the motor at physiological conditions.

In addition, the estimated K_d of ATP for a myosin bound to KNX-002 (0.7 μ M) shown in the Table in Figure 4c indicates that it is likely that under physiological ATP concentrations, the KNX-002 PfMyoA structure will also have ATP bound in the active site.

2- The experiments in Supp Figure 1 measure the affinity of Knx002 to apo and MgADP-myosin by quantifying the fraction of a slow kinetic phase of the MantATP binding transient. The finding that there are two resolvable kinetic phases suggests that the off rate of Knx002 is very slow. This is a point worth mentioning in the text. Also, is it possible that the affinities measured in Supp Figure 1 panels (a) and (b) are both measuring the affinities of Knx002 for apo myosin, i.e., the MgADP dynamically associates, so Knx002 can bind the apo state. If the inhibitor binding precedes nucleotide binding, then one might expect the affinities in the two experiments to be nearly the same.

As suggested, we added to the legend of Supplementary Figure 1 that the two resolvable kinetic phases suggest that the off rate of Knx002 is very slow. The text is now as follows (l. 28, Supplementary data):

“The two resolvable kinetic phases suggest that the off rate of KNX-002 is very slow.”

It is unlikely that the affinities measured in Supplementary Fig 1 panels (a) and (b) are both measuring KNX-002 binding to the apo state, because the observed time courses and rate constants differ for the apo conditions and the MgADP conditions (where saturating MgADP concentrations were used) while the amplitudes are similar. If we were observing KNX-002 binding only to the apo state (which would be a very small population in the presence of ADP), then we would expect the same rates whether ADP is present or not, and a smaller signal amplitude in the presence of ADP. Moreover, we know from our crystallographic data that KNX-002 is compatible with the ADP bound

state. Lastly, as mentioned above, we also performed an experiment where we mixed 200 μM KNX-002 with PfMyoA.mant-ADP and observed a small increase in bound mant-ADP fluorescence at a rate of $\sim 1 \text{ s}^{-1}$ ($n=2$), consistent with the ability of KNX-002 to bind to PfMyoA that has mant-ADP bound at the active site. This signal must come from bound nucleotide because we excite the mant-nucleotide via energy transfer from a Trp residue of PfMyoA. This new experiment is described in the last sentence of the legend of Supplementary Figure 1b (l. 40-43, Supplementary data).

In a separate experiment, when 200 μM KNX-002 was mixed with PfMyoA.mant-ADP, a small increase in bound mant-ADP fluorescence (excited via a Trp residue in PfMyoA) was observed at a rate of $\sim 1 \text{ s}^{-1}$ ($n=2$). This observation is consistent with the ability of KNX-002 to bind to PfMyoA that has mant-ADP bound at the active site.

Toward the goal of measuring these values more directly, we have also attempted to measure the affinity of KNX-002 to apo and Mg.ADP-PfMyoA by ITC, but ran into complications arising from the relative low affinity of this inhibitor for myosin. Buffer mismatches caused by excess DMSO needed to achieve high KNX-002 concentrations prevented us from measuring these values by ITC.

3- The rate of the slow phase in Supp Fig. 1 (a) is reported to be 1/s. According to the rate constant reported in Fig. 4, it should be 1.6/s. There are no uncertainties in Fig. 4, so I don't know if this is within the experimental error. Alternatively, it is possible that the curve in Fig. 4 "plateaus" at higher concentrations. Please comment.

The concentrations stated in the legend of Supplementary Fig 1 are the pre-mixing concentration, and we now also state the final concentration after mixing (i.e. 2-fold lower) in the legend. "(after mixing concentrations are 1.5 μM mant-ATP or 3 μM mant-ADP)." (l. 21-22, Supplementary data). From Figure 4 of the original main text, the calculated rate of the slow phase at 1.5 μM mant-ATP (not 3 μM mant-ATP) would be 1.2/s, in closer agreement with the value of 1.03/s stated in Supplementary Figure 1.

Having said this, since the original submission, we have repeated the mant-nucleotide binding data with newly purchased nucleotide stocks and expanded the range of nucleotide concentrations measured (1-12 μM) to obtain more accurate association and dissociation constants (now presented in a new Figure 4c). The rate constant in new Figure 4c does not align with the rate constant value from Supplementary Fig 1. However, as we are only monitoring changes in amplitudes as a function of KNX-002 concentration in the Supplementary Figure 1 experiment, the precise rate constant used for the fits to the fast and slow phases will have little effect on the reported KNX-002 affinity that is based on changing amplitudes of the two phases as a function of KNX-002 concentration.

4- I greatly appreciate the details provided in Supp Figs. 3 and 4 and in Supp Table 1. They are useful for understanding the structures and their relationships to other myosin isoforms. However, I think it would be appropriate for the authors to compare the residues required for Knx002 binding to other members of the myosin superfamily. It is not surprising that the myoA Knx002 binding pocket is very different from class-2 myosins. It would be of additional interest and importance to compare the sequences to unconventional myosins that show more homology to myoA. I did a brief survey of sequences, and myoA does seem to be divergent at these sites – so it would give the reader confidence in the selectivity of this inhibitor.

As requested, we now provide an alignment of some human unconventional myosins with PfMyoA in **Supplementary Figure 4b** to show how the residues that interact with KNX-002 differ between these myosins.

5- I was not able to get the Supp movies to open on my Mac. I did not try a Windows computer.

The format of the movies has been changed. They are now in .mp4, which is suitable with all the players (Windows Media Player, VLC). In our hands, the movies have played on various computers.

Reviewer #2 (Remarks to the Author):

This is an important piece of work, describing for the first time a lead compound specifically targeting Plasmodium falciparum myosin A (MyoA), which is the key component and motor of the molecular complex responsible for generating force for the malaria parasite gliding motility and host cell invasion. The manuscript describes a new compound, Knx002, that specifically inhibits the basal and actin-activated ATPase activity of MyoA, having no effect on cardiac or skeletal Myo2. Two crystal structures of MyoA at resolutions of 2.0 and 2.1 Å in the apo-ADP form and in complex with the inhibitor Knx002 and the poorly hydrolyzable ATP analog ATP-gamma-S, respectively, are presented and analyzed. In addition, effects of Knx002 on the affinity and kinetics of MyoA binding to nucleotides and actin are reported. The work should be published and will have an impact on the field and also open possibilities for drug design against an important global health threat. However, the manuscript needs additional work and some statements/conclusions should be either better backed-up, toned down, or reconsidered before I could recommend its publication.

We thank Reviewer 2 for the positive comments and helpful suggestions for revision.

I will list my specific comments below, divided into major, minor or more technical, and typographic/linguistic/stylistic points.

Major points:

- The authors spend a lot of words describing the novelty of the Knx002 binding pocket in MyoA. The question is: Can the pocket really be considered “novel”? It seems from the figures and the residues involved in the binding site that it is very much the same binding site previously described for blebbistatin, albeit the binding mode is different. The statements on the “novel binding site” should be toned down or the authors should convince the reader how the pocket, not just the binding mode, is new.

We agree with the reviewer that we could have made clearer why the pockets should be considered different and thus why KNX-002 has a novel binding site. We have now changed the figures and included a movie (**Supplementary Movie 3**) to make this clear. Supplementary Table 3 was previously written for this purpose but it was mistakenly not mentioned in the text. We now direct the reader to this table in the text so that this information is highlighted and can readily be referred to. Overall, it is important to understand that *in silico* docking of KNX-002 in a ‘blebbistatin’ pocket would absolutely not be able to provide a good model of the interaction (and vice-versa) mostly because there are large changes in the inner pocket of myosin between Post-Rigor and Pre-powerstroke states which thus define very different ligand binding pockets.

Blebb and KNX-002 are thus considered different for two main reasons. **First**, only few of the residues that bind Blebb and KNX-002 are similar (see **Figures 3b, 3e, 3f**, in which additional elements were added to make this statement clearer), the two compounds have a different scaffold and occupy their pocket differently with a shift in position (compare **Figure 3a, 3d** and new **Supp Movie 3**). The

polar and π -stacking interactions are of different nature for the two pockets. **Second**, the drug pockets are described as “cryptic” in myosins since due to conformational changes, these pockets are (i) not open during the entire cycle and (ii) do not display the same conformation along the cycle (see Planelles-Herrero *et al.*, 2017). This is why KNX-002 and Blebb do not target the same state (PR and PPS, respectively), but also why each pocket would not be compatible with the other compound: Blebb could not bind in PR and KNX-002 could not bind in PPS. Moreover, due to the different scaffold of the two drugs, the SAR would differ.

For these reasons, and because a pocket is defined by its volume, shape and flexibility that all contribute to drug binding, we conclude that the two pockets are different. This was stated in the text in paragraph 4 of the section entitled “KNX-002 binds in a previously undescribed pocket”.

To clarify for the reader and better illustrate the difference between the structures of myosin bound to these two different inhibitors, we have added a movie (**Supplementary Movie 3**) that compares KNX-002 with Blebb. The movie also zooms into the pockets to visualize the differences. We also revised **Figure 3** and its legend and we added in the text a reference to **Supplementary Table 3** that lists the similarities and differences between the interactions established with KNX-002 and Blebb. To better illustrate this point to the reader, a superimposition of PfMyoA/ATP γ S/KNX-002 and DdMyo2/Blebb was shown Fig. 3f, showing without ambiguity that (i) the elements within the two pockets have dramatically different conformations and that (ii) KNX-002 would not fit in Blebb pocket and that its presence would induce some clashes with Switch-2.

- The mechanism of inhibition of ATP hydrolysis is stated to be the impact of the compound on the Mg²⁺ coordination. Although the data do suggest that as a possibility, it is not possible to come to such a conclusion without presenting a catalytic mechanism and having structures of MyoA in both ATP (or ATP-gamma-S) and ADP form in the presence and absence of the compound. Comparing apo-ADP-MyoA and Knx002-ATP-gamma-S-MyoA and ATP, ATP-gamma-S, and ADP structures of another myosin (Myo2) and the simulations reported are not direct comparison that can be used to firmly draw this conclusion. Either the authors need to perform additional experiments to confirm their hypothesis and propose a detailed catalytic and inhibition mechanism or the conclusion has to be toned down. Because Knx002 inhibits hydrolysis, it should also be possible to get an ATP structure with the compound.

We thank the reviewer for his/her suggestions and we indeed removed the statement regarding the role of the Mg²⁺ ion in the KNX-002 mechanism. We now discuss possible reasons why KNX-002 can slow hydrolysis of ATP, without mentioning Mg²⁺ coordination.

Indeed, additional structures were obtained to visualize different ATP analogs bound in the active site. For the ATP analog that is hydrolysable (ATP γ S) and which is arguably the closest to ATP, structures with KNX-002 present or not showed that the Mg²⁺ ion is in fact hexa-coordinated in a similar way as previously seen for structures of other myosins bound to ATP analogs. Interestingly, however, Mg²⁺ coordination is perturbed in KNX-002 structures bound to ADP.BeFx. In this case, the position of the Mg²⁺ was without ambiguity displaced as illustrated in the previous Figure 4 of the original manuscript. Since historically, the ADP.BeFx structure was the first structure we determined and since it was solved without ambiguity at high resolution (~ 2.2 Å resolution), we had first presumed that the Mg²⁺ coordination would be also perturbed for all ATP analogs and for ATP itself. After collection of several ATP γ S datasets with KNX-002 co-crystallized, we now have no doubt that ATP γ S allows hexa-coordination of Mg²⁺.

Additional crystal structures at lower resolution (~3.1 Å) suggest that the Mg²⁺ coordination is also hexa-coordinated when ATP is bound. We obtained these structures by soaking ATP in nucleotide-free, KNX-002 bound crystals. Note that co-crystallization requires a long-time scale for crystals to appear (more than a week) and requires the protein to be in equilibrium between a pool free in solution and a pool in the crystal. Under these conditions, one cannot prevent ATP hydrolysis by the free pool in solution and the crystals contain ADP instead of ATP in the active site, as the small amount that remains in the apo form in solution is present in sufficient amounts to hydrolyze all ATP over time.

Since BeFx is a poorer γ -P_i mimic compared to that found in ATP or ATP γ S, we no longer think that Mg²⁺ plays a role in inhibition under physiological conditions. All structures confirm that KNX-002 binding does not favor ATP hydrolysis, but they also reveal that the mechanism inferred from ADP.BeFx is not correct and should be dismissed. ATP binding in fact favors a hexa-coordinated coordination as seen in the structures that do not have KNX-002 bound for either PfMyoA or other myosins.

We have modified the section “**Characterization of the impact of KNX-002 on the nucleotide binding site**” to include these additional results and we have modified Figure 4 and Supplementary Figure 6, accordingly.

“The fact that both ATP γ S and KNX-002 are compatible in the same structure and their mutual presence does not lead to hydrolysis of ATP γ S indicates that KNX-002 prevents hydrolysis, most likely by perturbing the ability to favorably position the water molecule required for hydrolysis.” (l. 325-328)

Our model of PfMyoA inhibition proposes that KNX-002 traps the motor in the post-rigor state as the presence of this ligand prevents rearrangements of the inner cleft required for the recovery stroke and adoption of the pre-powerstroke state by the motor. In addition to the fact that the post-rigor state is not able to hydrolyze ATP efficiently (Bauer *et al.*, 2000), our structures suggest that the alteration of the coordination near the gamma-phosphate would increase the difficulty of hydrolyzing ATP since the catalytic water would not be in place to favor hydrolysis. The results section now reads as follow in the section “**Characterization of the impact of KNX-002 on the nucleotide binding site**”:

In particular, we now include the following:

“Taken together, these results suggest that PfMyoA adopts a stable PR state when both ATP and KNX-002 are bound, in which the inner cleft cannot close, thus preventing the recovery stroke. The water near the pyrazole, observed in our PfMyoA-MgATP γ S structure bound to KNX-002, is not in position to favor attack of the γ -P_i of ATP. The fact that both ATP γ S and KNX-002 are compatible in the same structure and their mutual presence does not lead to hydrolysis of ATP γ S indicates that KNX-002 prevents hydrolysis, most likely by perturbing the ability to favorably position the water molecule required for hydrolysis. The observation that KNX-002 separates the active site from Switch-2Glu474 and slows Switch-2 rearrangements excludes a ATP hydrolysis mechanism that normally occurs in myosins, which requires the repositioning of Switch-2 during the recovery stroke so that the _{switch-2}glutamate can promote ATP hydrolysis by coordinating the attacking water molecule (Kiani & Fischer 2015; Chakraborti et al 2021). Our structures thus provide a mechanistic explanation for how KNX-002 disfavors ATP hydrolysis.” (l. 322-332)

Part of the discussion (paragraph 3) has been edited as follow:

“Our structure suggests that the molecular basis for reduced ATP hydrolysis includes slowing the recovery stroke and the inability of the motor to adopt the pre-powerstroke state, which is the most favorable to promote ATP hydrolysis by positioning a water molecule for in line attack of the γ -phosphate. Blebb and KNX-002 therefore belong to different categories of inhibitors with different mechanisms of action. KNX-002 is the first representative of a novel class of myosin inhibitors, whose mode of action results from sequestering a post-rigor state, that is incompetent for ATP hydrolysis. Trapping PfMyoA in a post-rigor state is sufficient to inhibit the activity since ATP hydrolysis does not occur in this state (Bauer et al., 2000).” (l. 409-416)

- *In all the actin-related assays, vertebrate skeletal muscle actin has been used. Why not use parasite actin, as the authors are in a unique position by having access to recombinant P. falciparum actin, on which they have published high-impact articles (e.g. Robert-Paganin et al. Nat Comm 2021; Lu et al. PNAS 2019)?*

In Robert-Paganin et al Nat Comm 2021 Table 1, we presented data showing that PfMyoA supports motility of PfAct1, skeletal muscle actin, or smooth muscle actin at indistinguishable rates, as a consequence of the surprisingly conserved core interface common to all actomyosin complexes, even those that are evolutionarily very distant. Based on these prior data, we used skeletal actin in the assays as large amounts of PfAct1 for these assays are not trivial to obtain. Moreover, it would be difficult to perform the ATPase assays with unstabilized PfAct1 filaments as the filaments would be changing with time. Once one adds jasplakinolide to stabilize the PfAct1 filaments they become much more like skeletal actin in their behavior.

- *It seems that it is not possible to draw conclusions on the in vitro motility assays if almost no filaments could be observed in the presence of the compound. This is both stated in the manuscript and visible in the two supplementary movies. Given this, it is strange that the effect of the compound on actin polymerization is not discussed and not characterized. This is a critical point, given that the assays were performed using skeletal muscle actin (see above), and it seems that the lead compound actually may inhibit host actin polymerization. Could this also be a reason for the toxicity of the compound in cell culture experiments (see below)?*

The reason that no actin filaments were observed is that we purposely reduced the methylcellulose concentration to 0.15% (as stated in Methods) so that actin filaments not tightly bound to myosin could diffuse away from the focal plane rather than be forced to engage with myosin (methylcellulose constrains actin diffusion). This led to the “all or none effect” as a function of KNX-002. If the methylcellulose concentration was increased to 0.5% (data we did not present), the actin filaments cannot diffuse away from the surface and slow motility was observed in the presence of KNX-002, likely powered by a few myosin heads without KNX-002 bound. In motility assays you can knock out the function of many heads with no effect on speed until you reach a critical concentration of motors where speed begins to decrease with myosin concentration. This depends on the duty ratio of the motor.

We clarified the sentence in the text

In addition, very few actin filaments were bound to surface-immobilized myosin in the presence of KNX-002 (6.5 ± 1.9 filaments/field versus 110.3 ± 1.9 filaments/field in its absence, $n=6$ fields). (l. 101-103) (l. 101-103)

Nonetheless, we performed a TIRF polymerization assay with PfAct1 in the presence or absence of 100 μ M KNX-002 to show that KNX-002 has no effect on PfAct1 filaments. This experiment is relevant for interpretation of the parasite growth assay.

We added the following sentence to the first paragraph of Results.

In vitro TIRF polymerization assays further showed that KNX-002 did not affect PfAct1 filament assembly, which polymerized to similar lengths and at the same rate in the absence (16.6 ± 2.3 subunits/s, $n=36$) or presence (17.2 ± 2.5 subunits/s; $p=0.28$) of KNX-002. (l. 103-106)

- *The compound has a low affinity and also significant toxicity at the high concentrations used in the assays. In the text, the authors quote survival rates of three different cell types (Supplemental Table 1) at a Knx002 concentration of 20 microM. At this concentration, the fibroblast and epithelial cells seem unaffected, but ~20% of the hepatic cells died. The assays have been performed at 100 or 200 microM concentrations. At 100 microM, only the epithelial cells remained unaffected, whereas ~20% of the fibroblasts and ~70% of the hepatic cells died. At 200 microM, basically all cells died (except for ~20% of the fibroblasts). Why are the cell survival numbers given for much lower concentrations than used in the assays and why is this toxicity not considered an issue? 100 and 200 microM concentrations are tens of times IC50. Were lower concentrations tested? What happens at IC50? Is there any measurable effect?*

KNX-002 is a first hit that could lead to the development of anti-malarial compounds. Kainomyx has in fact developed a more potent compound based on the KNX-002 scaffold and has introduced these compounds in different cellular assays to assess the potential of the series (Trivedi et al., 2022; <https://doi.org/10.1101/2022.09.09.507317>). As our paper mostly describes the pocket in which such a compound can bind and how these compounds could mechanistically impact motor activity, we decided to cite the experiments on toxicity that were reported for KNX-002 rather than using our own data. We refer in the Discussion to a more comprehensive study of KNX-002 toxicity performed by Ward and colleagues (<https://doi.org/10.1101/2022.09.09.507210>). The text in Discussion is now as follows:

“Moreover, KNX-002 was shown to not be toxic to human HepG2 liver cells at 80 μ M (Kelsen et al doi: <https://doi.org/10.1101/2022.09.09.507210>).” (l. 421-422)

Note that difference in the methodology for these assays as well as the sensitivity for the cancer cell line Hep-G2 that we had used probably explains the difference between the Ward study and our study. For toxicity, the relevant cell lines should be cancer-free and KNX-002 is shown to have low impact on the non-cancerous cell lines.

- *The triple mutant has a large effect on the sensitivity of the mutant to the compound in the ATPase assay. Were the effects of these mutations on the folding and stability of the protein assessed in any way? It would be also very informative to see the active site configuration of this mutant protein.*

To assess the integrity of the triple mutant PfMyoA, we showed that it moved actin smoothly in an *in vitro* motility assay with speeds that fit a Gaussian distribution both in the absence or presence of KNX-002. SDS-gel analysis also showed an intact heavy chain. These data are presented in a new **Supplementary Figure 5 - Characterization of the triple mutant F270Y/F471A/F645H**. Reference to this supplementary figure has been added in the text, in the last paragraph of the section “KNX-002 binds in a previously undescribed pocket”:

*“The triple mutant showed intact heavy and light chains on an SDS-gel, and its functionality was assessed by an *in vitro* motility assay where it moved actin filaments with a Gaussian distribution of speeds that was the same in the presence or absence of KNX-002 (Supplementary Figure 5). This result is consistent with the binding pocket of KNX-002 identified crystallographically, and also*

suggests that some or all of these three key-aromatic positions are essential in the efficient binding and specificity of KNX-002 for PfMyoA.” (l. 275-280)

- This may already be out of the scope of this work, but of course ideally, compounds like this should also be evaluated in a mouse model.

This is indeed out of scope of this work given that this compound will not be developed by us but by Kainomyx who has submitted a BioRxiv paper (<https://doi.org/10.1101/2022.09.09.507317>) describing a more potent inhibitor based on the KNX-002 scaffold. They will likely perform the experiments in the mouse. This work is now cited in the final paragraph of Discussions in order to address this point and strengthen the potential of KNX-002 SAR approaches:

“A more potent derivative based on the KNX-002 scaffold (KNX-115) was recently characterized *in vitro* and in the *Plasmodium* parasite by James Spudich (CEO of Kainomyx) and colleagues (Trivedi *et al.*, 2022). KNX-115 shows great promise as a therapeutic agent because it is parasitocidal at multiple stages of the *Plasmodium* lifecycle, acts on resistant *Plasmodium* strains and displays no liver cell toxicity (Trivedi *et al.*, 2022). (l. 427-431)

Minor or technical points:

- Why are the diffraction data so incomplete? Because of anisotropy? This can have an impact on the quality of the ED maps. Low completeness in the low resolution shells can cause distorted maps or missing parts in the electron density. Given this and without having access to the data or the maps, it is hard to say whether it can with certainty be said that the apo structure has ADP and not disordered ATP-gamma-S. The completeness of the data sets (in Supplementary Table 2) is given as spherical completeness and with ellipsoidal correction. However, it is not clear if the rest of the statistics given are for the non-corrected or corrected data? What correction method was used? Which data were used for refinement and map calculation? Did the ellipsoidal correction have an impact on map quality - could important features be seen in both maps?

The crystals are difficult to grow and to handle. They take more than a week. The datasets we selected however, provide good maps that we can provide to the reviewer for his assessment. We also provide snapshots of the density in the active site below for describing why we are confident that there is no doubt on the fact that ATPgS was hydrolysed in the experiment without KNX-002 but not in the similar experiment with KNX-002 present.

The questions raised by Reviewer 1 (question 1 on whether the drug can bind to its site when nucleotides are present) and by this reviewer (to evaluate whether we could see ATP or other nucleotide bound in the active site) led us to collect more data sets. While each type of crystal ended up showing the same features, one crystal with ATPyS and KNX-002 led to a better dataset that is now included in the manuscript.

As stated in methods, we used the software STARANISO to apply the “anisotropic correction” (Tickle *et al.*, 2016). The major strength of this method is to use locally averaged values of $I/\sigma(I)$ (signal/noise) instead of spherically averaged ones, classically used. Then it allows to process data by regions and to take into account the anisotropy.

The STARANISO software process data with the following steps. (i) It determines an anisotropic diffraction cut-off of the merged intensities using the locally averaged mean $I/\sigma(I)$. (ii) Systematic

absence factors combining likelihood function (Popov & Bourenkov, 2003) and Bayesian approach (French & Wilson, 1978) are used to evaluate the anisotropy. The method allows thus to get maximal information from a dataset by processing the anisotropic regions. Concerning the completeness, two are provided. **(1) Spherical completeness** which is calculated classically as in isotropic processing. **(2) Ellipsoidal completeness** which is calculated conventionally as the fraction of reflections inside the data-dependent cut-off surface that were measured, in this case the geometry is elliptical and defined by the above method.

Given the statistical robustness of the method, there is no bias to fear from it. The delicate step is to be sure about the anisotropic cut-off and to be sure that it will not introduce noise. For both datasets published in this work, the quality of the statistics and of the maps are excellent. There is no doubt about the presence of the ligand and no ambiguity about the ADP as it can be seen in the PDB validation reports (section "ligands"). **In order to help Reviewer #2 to evaluate the quality of the structures, we are happy to provide the models and the maps in the revised version.**

It is true that anisotropy may be a problem for the quality of the crystals and specifically for the visualization and building of the ligands. In the present case, we had no ambiguity for positioning of the ligands. We collected new data from co-crystallization experiments that have both KNX-002 and ATP γ S. We now selected the best data set of this series, the Mg²⁺ ion and its coordination are rebuilt without ambiguity from the Fo-Fc peaks in the electron density maps (**Rebuttal Figure 1a**). Zooms of the density for the Mg²⁺ or the γ -P_i in the 2Fo-Fc map demonstrate that there is no ambiguity about the coordination of the Mg²⁺ ion or the presence of the third phosphate in the density (**Rebuttal Figures 1b, 1c**). Crystals were also obtained from similar conditions by co-crystallization with ATP- γ S without KNX-002 (**Apo condition**). In this case, the nucleotide and the Mg²⁺ ion are visualized without ambiguity from the Fo-Fc map (**Rebuttal Figure 2a**). Model building and refinement show the presence of an ADP and a hexa-coordinated Mg²⁺ ion without ambiguity (**Rebuttal Figure 2b**). The absence of electron density indicates the absence of γ -P_i (**Rebuttal Figure 2b**). We can thus claim that there is no ambiguity about the fact that the nucleotide bound is hydrolyzed in the absence of KNX-002 but not in its presence and we can describe the coordination of the Mg²⁺ ion in the two datasets without ambiguity.

Rebuttal Figure 1: Nucleotide and ligands in the PfMyoA-KNX-002-ATP γ S crystal. (a) displays the Fo-Fc map peaks of the ligands. The ligands are seen without ambiguity in the difference map: KNX-002 (red); ATP γ S (white), magnesium ion and coordination (yellow). (b) Shows the rebuilt magnesium and coordinating waters (yellow) and the γ -phosphate of ATP γ S (white). Both the hexa-coordination of the Mg²⁺ ion and the presence of the γ -phosphate are visualized without ambiguity in the 2Fo-Fc electron density map. (c) shows the rebuilt KNX-002 in the 2Fo-Fc map (in red). The 2Fo-Fc map is contoured at 1.5 σ and the Fo-Fc map is contoured at 3 σ .

Rebuttal Figure 2: Nucleotide and ligands in the PfMyoA-Apo-ATP γ S crystal. (a) displays the Fo-Fc map peaks of the ligands. The ligands are seen without ambiguity in the difference map: ADP (white), Mg²⁺ ion and coordination (yellow). (b) Shows the rebuilt Mg²⁺ ion and coordinating waters (yellow). Note the absence of density for the γ -phosphate of ATP γ S (circled in white dotted lines). Both the hexa-coordination of the Mg²⁺ and the absence of the γ -phosphate are visualized without ambiguity in the 2Fo-Fc electron density map. The 2Fo-Fc map is contoured at 1.5 σ and the Fo-Fc map is contoured at 3 σ .

- Why is the number of replicates only 2 in many of the experiments? Usually, triplicate series should be expected.

When only 2 replicates are shown, there was very little variance between our 2 data sets making the need for and the value of a third replicate data set less important to the conclusions of the paper. In Figs 4 and 5 the n=2 represents data obtained with independent protein preparations which is fairly stringent.

- What is n in Figure 1c?

We added that n=2.

- The cell survival assay methods are not described at all.

This was an oversight, sorry. We deleted our experiments on toxicity, and instead referred in the last paragraph of the Discussion to a more comprehensive study of KNX-002 toxicity performed by Ward and colleagues (<https://doi.org/10.1101/2022.09.09.507210>).

- The purification of cardiac and skeletal myosins are not described. The purification of skeletal muscle actin is described under the title “Myosin expression and purification”. The title should be “Protein expression and purification”, and those should be described for all the proteins used.

We changed the title of this Methods section to :“Protein expression and purification” and added a reference for skeletal and cardiac myosin purification.

- The description of the phosphate burst assay is minimal. Either references to literature should be provided or the method described in sufficient detail, allowing the reader to understand exactly how it was done. How was the possibility of the very common PO₄ contamination from purification and reagents taken into account?

The paragraph in Materials and methods has been revised to read:

“Chemical hydrolysis of ATP by myosin was performed by manual mixing under single turnover conditions. PfMyoA (25 μM) was pre incubated with 1% DMSO or Knx002 (100 μM in 1% DMSO), manually mixed with 20 μM ATP, aged for 5 s, and then quenched by addition of 0.3 M perchloric acid before quantifying free phosphate using malachite green (Fisher) by the method described in ³⁸ which utilizes a phosphate standard curve to correlate OD595 signal change to nmoles phosphate. Controls using myosin, dialysis buffer or ATP alone showed undetectable phosphate. Conditions: 10 mM imidazole pH 7.5, 150 mM KCl, 4 mM MgCl₂, 1 mM EGTA, 10 mM DTT, 25 μM PfMyoA, 20 μM MgATP, 1% DMSO, 30°C.” (l. 541-548)

- *In the transient kinetics assays, 3-8 traces were measured. A common strategy is to measure at least 10 traces, so that there still remains a large enough number of observations in case some need to be excluded as outliers. Would 3 traces mean that most of the data were excluded as outliers?*

In general fewer traces were measured when the data collected were superimposable, representing the lower limit stated. More traces were obtained/measured when some variance was observed between traces allowing outliers to be excluded from the final averaged data traces that were then fit to one or more exponentials using KinTek software.

- *Also, in the transient kinetics assay, the protein concentrations were fairly low. Possibly this is not a problem with the fluorescent nucleotide analogs, but why is the ratio of actin:myosin not 1:1? In Figure 4a, there are no error bars or standard deviations given.*

We designed the experiments with a ratio of actin:myosin greater than 1 (1.25 molar excess of actin over myosin) to ensure that all myosin in solution will be bound to actin and there is no chance of having a population of free myosin. The error bars for Figure 4c are within the symbol size and are thus not visualized (now stated in legend).

- *The simulations are poorly described. It seems, based on the methods description, that the Mg²⁺ ion was placed in the “expected” position from the ADP structure at the start of the simulations. At the timescale used (220 ns), it is probably not to be expected that it would move, and the side chains would be more*

likely to move to accommodate it. With today's computing power, simulations in the ms scale are not out of reach. Can the used force field handle divalent cations, which have been notoriously challenging for simulations?

The simulations have been removed from the manuscript as we no longer discuss the unconventional position of the Mg²⁺.

Note however, that the protocol used was classical for these simulations. We used it in several of our past publications (see Robert-Paganin et al., 2018; Robert-Paganin et al., 2019; Moussaoui et al., 2020; Robert-Paganin et al., 2021). Hundreds of ns are indeed sufficient to observe movements in subdomains such as Converter orientation (see Robert-Paganin et al., 2019).

- In Supplementary Figure 1, it seems that there are no error bars in panel (a). Are they missing or are the errors so small that they are invisible? This is one of the experiments, where only duplicates were performed/used.

Supplementary Figure 1A shows two experiments with slightly different conditions as detailed in the legend. Supplementary Figure 1B is a duplicate experiment where error bars are shown. As now stated in legend, some error bars cannot be visualized as they fall within the size of the symbol.

- In the text, U50 and L50 should be explained to readers not so familiar with myosin structure.

We have changed the text as follows in the **'KNX-002 binds in a previously undescribed pocket' section:**

"In the PfMyoA/ATPγS/KNX-002 structure, KNX-002 is buried in a tight cryptic pocket located between the so-called Upper 50 kDa (U50) and Lower 50 kDa (L50) subdomains of the motor domain that both have elements that bind the actin filament. KNX-002 binds in the "inner cleft"..." (l. 182-184)

This is also introduced in Figure 2 legend.

- In the text: "Most compounds to date target the PPS state" - I suppose this refers to myosin inhibitors in general? It would be good to state this more clearly.

We have changed the text in the section entitled "KNX-002 binds in a previously undescribed pocket" as follows:

"Most myosin inhibitors described to date target the PPS state of the myosin cycle (Supplementary Figure 2)^{14,27,16}." (l. 190-191)

- The final concentration of DMSO in the assays is not always very clearly stated. Is it always 1% or below, as 1% was used in the control experiments?

DMSO is always 1%, whether inhibitor was present or not. This is stated in Methods where buffer components are explicitly given.

Typographic, linguistic, stylistic, etc. issues:

- The authors refer to MyoA as "atypical". This might suggest that MyoA is not a typical myosin of its class (class XIV). Unconventional would seem like a better choice of wording.

We changed atypical to “class XIV” in the abstract (l. 45).

- The sentence: “627,000 people died of malaria in 2020, the majority being children under the age of 5 years”. Better would be not to start a sentence with a number.

Sentence now reads “In 2020, malaria was responsible for 627,000 deaths, the majority being children under the age of 5 years.” (l. 58-60).

- In the methods section “Myosin expression and purification”: What are PUNC chaperones? Should it be “PfUNC”? In any case, it should be spelled out.

The sentence now reads:

“The full length PfMyoA heavy chain WT and mutants were co-expressed with a UCS (UNC-45/CRO1/She4p) family myosin chaperone from *Plasmodium* spp. and two light chains PfELC and PfMTIP (Bookwalter *et al.*, 2017).” (l. 500-502)

- In the methods section “Myosin expression and purification: “constructs were purified”... “Construct” usually refers to the DNA construct encoding the protein. This should be reworded.

We have deleted the sentence “The constructs were purified for the crystallization assays.” And we include now “The FLAG-tagged recombinant proteins were purified from baculovirus-infected Sf9 cells using previously described methods in³⁶.” (l. 502-503)

The other sentence in Methods in the “Crystallization, data processing...” Section now reads : “Full-length PfMyoA with bound light chains PfELC and MTIP-ΔN (lacking residues 1-E60) was co-crystallized with or without KNX-002.” (l. 567-568)

- Figure 1b is strange and not explained well in the figure legend.

The figure has been edited for clarity. The size of KNX-002 has been increased and the parasite cycle removed.

- There is a dot missing after “Knx002” on line 4 of the legend of Figure 2.

Added

- Figures 6 and S2 seem very similar and to a large part redundant. Figure S2 probably was included so that it can be referred to early in the text, while still having Figure 6 as the last one. It might be worth considering making these more different from each other.

We considered this suggestion, but feel that keeping similar figures is easier for the reader that would not be familiar with the myosin motor cycle. We thus decided to keep it since we are referring to structural states at several places in the manuscript. The main difference between these figures is that one introduces the cycle for readers while the other provides a graphic description of the mode of action of the KNX-002 inhibitor and contrasts it to Blebbistatin and MPH-220 inhibitors.

- The colors in Figures 2, 3, and 5 could be improved. The colors, especially the blue and green/cyan shades, are difficult to tell apart. It would be better to use colors that differ more clearly from each other. Some of the color names/codes used in the figure legends are not real color names but rather codes used in the

programs used to make the figures. It would be better to use e.g. “green” instead of “deep teal cyan”, “blue” (or “dark blue”) instead of “marine blue”, “beige” instead of “wheat”, etc.

These color names are those from pymol, the software with which we have done the figures. By convention and in order to allow reproducibility of our color code convention by the reader, we chose to use these names.

- On line 11 of the legend of Figure 3, “conserved residues” should be in singular, as only one residue is in a purple box.

This has been changed.

- On lines 15-16 of the legend of Figure 3, the word “binding” seems to be missing after “Blebb”.

The word “binding” has been added.

- On the 9th line of the legend of Figure 5, the word “(colours)” probably should be replaced by the actual colors used in the figure.

This has been changed.

- The style of the figures in general could be improved and unified; Some of the panels are in boxes, some not, the order of the panels is not always logical, there’s a mixed use of bold and normal fonts in the figures, some figures have text elements which are on the border of being too small to read.

This has been changed. The size of KNX-002 has been increased in Figure 1c and the parasite cycle removed. The space between the panels of Figure 2 has been increased for clarity. The font size in Figure 3, Figure 6 and Supplementary Figure 2 has been increased in size for readability. Figure 4 has been changed. Some small mistakes in Figure 6 have been corrected.

- It seems that Supplementary Table 4 is not referred to in the text.

We thank the reviewer for pointing this out. It is now added in discussion – note that Supplementary Table 4 became Supplementary Table 3:

“Interestingly, transient kinetics and structural data both demonstrate that the mechanism of action of these inhibitors greatly differ (Supplementary Table 3).”(l. 403-404-)

- The accuracy of the rmsd values for the superposition of the structures with three decimals seems a bit exaggerated. Also, the value given is not exactly the same in the main text and the figure legend.

This has been changed.

Reviewer #3 (Remarks to the Author):

In this study, the Authors characterize in detail a mechanism of action of a previously identified inhibitor of myosin A from Plasmodium (PfMyoA). PfMyoA is part of glideosome that is critical for the parasite mobility and infectivity, and is a validated drug target. The Authors combine multiple experimental and computational methods to characterize the selected compound action from the level of protein structure

to the parasite level. In my opinion, this is a very convincing, coherent study, which is additionally clearly written. The conclusions are consistent with the results. The results seem very useful for anti-Plasmodium structure-based **drug design**. I would recommend to publish this work after addressing the remarks below.

We thank Reviewer 3 for the positive comments and helpful suggestions.

Major remarks:

the "Methods" section:

It would be good if the Authors provided more information about the system setup for MD simulations:

- l. 527-528: "Starting from the PfMyoA/Apo and PfMyoA/Knx002 coordinates, ATP was modelled after ATP γ S and the Mg $^{2+}$ was positioned as found in PfMyoA/Apo."
- Could you provide more details how Mg $^{2+}$ was modelled in the liganded system? How well the surrounding of Mg $^{2+}$ can be aligned in the two systems? Maybe the alignment could be shown in SI?
- How Mg $^{2+}$ and ATP were parametrized for MD? (which parameters?)
- Was the hydration shell of Mg $^{2+}$ from the X-ray preserved in the built MD systems?
- How large were the systems? (no. of atoms in total and waters)
- What was the simulation protocol?

We would like to answer the reviewer's methodological questions below – although we have now removed the simulations from this manuscript due to the fact that we no longer think that when ATP is bound, the Mg $^{2+}$ coordination differs from that usually observed in the absence of compound, after acquisition of additional structural and functional data, as explained in the paragraphs addressed to the editor and the answers to the other reviewers. We have thus fully revisited the description of the mechanism by which ATP hydrolysis would be inhibited and the role of Mg $^{2+}$ is no longer an issue.

The protocol used for our simulations is classical for dynamics. We provide some details below although this is no longer in the paper:

"Starting from the PfMyoA/Apo and PfMyoA/KNX-002 coordinates, ATP was modelled after ATP γ S and the Mg $^{2+}$ was positioned as found in PfMyoA/Apo. All molecular dynamics simulations were performed with Gromacs 2018.3⁴⁸ on all-atom systems parametrized with charmm36m forcefield and built with CHARMM-GUI server⁴⁹. The box consisted in a cube of 149 Å as a length of the edge and a volume of 3307949 Å³. All systems consisted of a box with 97834 explicit water (TIP3) molecules and neutralized with salt (KCl reaching 150 mM). The pH of the system was set to 7.0 with no protonation of His residues. The Mg $^{2+}$ ion and waters part of the coordination were placed before minimization as visualized in the structure. Long-range electrostatic interactions were handled using the particle mesh Ewald (PME) method⁵⁰. The simulations were performed in an NPT ensemble; the temperature and the pressure of the system were fixed at 310.15 K with the Nosé-Hoover thermostat and 1 bar with the Parrinello-Rahman barostat. Trajectories of 220 ns were generated and further analyzed with the Gromacs tools and visualized in PyMOL⁵¹ which served to create the illustrations. Atomic displacements were computed with VMD (Humphrey *et al.*, 1996)."

Minor remarks:

- In my opinion, it would be good if the Authors introduced blebbistatin in the Introduction, since it is one of the main compounds analyzed (why is it used as a reference?).

We introduced a sentence in the introduction as requested.

From Structure Activity Relationship (SAR) on the Blebbistatin (Blebb) scaffold, MPH-220 was developed as a specific inhibitor of skeletal muscle myosin (SkMyo2) with promising therapeutical value against muscular spasticity¹⁶. (l. 77-79)

- the sentence l. 135-138 is too long and unclear:

"Whether bound to Knx002 or not, PfMyoA crystallized in the post-rigor (PR) state, an ATP-bound myosin structural state with low affinity for the actin track which is populated upon detachment of the motor from the track prior to the priming of its lever arm (Supplementary Figure 2)."

The sentence in the "**KNX-002 targets the post-rigor state**" paragraph was changed as follows :

Text in the previous version of the manuscript:

The high-resolution electron density maps are at similar resolution in these two datasets and allow us to build the nucleotides and Knx002 without ambiguity, as well as the water molecules, in particular those in the active site (Figures 2a, 2b). Whether bound to Knx002 or not, PfMyoA crystallized in the post-rigor (PR) state, an ATP-bound myosin structural state with low affinity for the actin track which is populated upon detachment of the motor from the track prior to the priming of its lever arm (Supplementary Figure 2). The two structures are highly superimposable (rmsd 0.214 Å on 914 Cα-atoms, Figure 2c), indicating that Knx002 does not induce major structural changes in the myosin structure.

New text now added to the manuscript:

The two datasets are both at high resolution and the resulting electron density maps allow us to determine that the PfMyoA structure adopts a post-rigor (PR) state, an ATP-bound myosin structural state with low affinity for the actin track that is populated upon detachment of the motor from the track prior to the priming of its lever arm (Supplementary Fig. 2). These high-resolution datasets also permitted us to position the nucleotides and KNX-002 without ambiguity, as well as the water molecules, in particular those in the active site (Fig. 2a, 2b). The two structures are highly superimposable (rmsd 0.2 Å on 914 Cα-atoms, Fig. 2c), indicating that Knx002 does not induce major structural changes in the myosin structure. (l. 149-156)

- l. 95-97: the sentences: "Because PfMyoA is essential for blood cell invasion 9,10, we tested the effect of Knx002 on P. falciparum asexual, blood-stage growth, itself dependent on the ability of merozoites to invade erythrocytes 23. Knx002 inhibited asexual blood stage growth of merozoites ..." are unclear to me. Could the Authors rephrase or expand these sentences?

The revised sentence reads:

"Because PfMyoA is essential for blood cell invasion^{9,10}, we tested the effect of KNX-002 on P. falciparum asexual, blood-stage growth, an assay dependent on the ability of merozoites to invade erythrocytes²³." (l. 114-116)

l. 174: "bonds" -> "interactions"?

This has been changed.

I. 248: Mant-ATP - for some readers it may be obvious, but the Authors could mention what is Mant-ATP and why it is used.

We added the phrase “...nucleotides whose fluorescence is enhanced when bound to PfMyoA.” (l. 292-293)

I. 335: what is "A.M.ADP"? This abbreviation seems not to be introduced.

We changed this to “actomyosin.ADP” (l. 371).

Figure 1:

(b) The dotted lines on the chemical structure are too thin to differentiate colors. The bottom part of the subfigure is not explained in the caption (I think the cycle should be briefly explained).

We have changed Figure 1 as proposed by the reviewer. The size of the panel 1b has been increased to improve the readability.

Figure 2:

There should be more space between the subfigures to make the figure more clear.

(c) The two structures should be shown in different colors, not colored by domain, because now conformational differences are hard to be seen or the subfigure should be skipped (RMSD would be enough). RMSD: it should be specified that it is C α RMSD.

We arranged the figure according to these requests. The central panel has been moved to improve the readability.

As expected with the rmsd of 0.2 Å, the two structures are very similar and it is expected that the differences cannot be easily seen. Figure 2c shows that the two molecules cannot be distinguished in this overall view due to their similarity and we mention this in the legend now. We also mention that only local rearrangements of side chains as seen in Figure 3c or Supplementary Figure 3a occur. These figures compare these structures and zoom on the region where the largest changes are found near where the drug binds. It shows that only local changes occur. By zooming near the nucleotide, these figures also show that only local changes are found between the KNX-002 and apo structures, when ATP γ S is bound in the active site.

We changed the legend to figure 2 to “*The compound does not induce major structural rearrangements upon binding. PfMyoA-KNX-002-ATP γ S and PfMyoA-Apo-ATP γ S are both in a PR state and superimpose quite well with a rmsd of 0.2 Å using the C α atoms. Zoom on the regions with maximum differences between the two structures show local displacements of side chains (see Figure 3c, Supplementary Fig. 3a).*” (l. 140-143)

RMSD were mentioned to be done with C α .

Figure 3:

(b) Are interaction identification criteria defined somewhere? Were the interaction diagrams generated by some software?

We added a section to the **Material and Methods** in order to address this point. The paragraph appears as follow:

Analysis of the drug binding pockets

The residues involved in drug binding were automatically determined with the software LigPlot+ (Laskowski & Swindells, 2011). Default set up interaction cut off at 3.9 Å, but we investigated manually longer range interactions (yet < 5 Å) by visualization with the pymol software (Schrödinger & DeLano, 2020). The figures panels were created and manually edited with LigPlot+. (l. 594-597)

(d) Closing bracket is missing.

Fixed, thank you

(f) "Residues involved in Blebb are circled" - the word "binding" is missing.

We thank Reviewer #3 for this remark. It has been fixed.

l. 375-377:

"Importantly, it is the first time that a compound is reported to bind to the inner pocket of a Post-rigor myosin state, without the requirement for much conformational change." - this sentence is a bit unclear: conformational change of what?

We meant to say : "without much requirement of conformational change from the apo post-rigor structure". We now have simplified the sentence (in Discussion) as the following statement is sufficient.

KNX-002 is the first representative of a novel class of myosin inhibitors whose mode of action results from sequestering a post-rigor state. (l. 413-414)

REVIEWERS' COMMENTS

Reviewer #1 (Remarks to the Author):

As stated in my original review, experiments in the paper are thoughtfully designed, and the paper is well written. The work is impactful and will be of interest to the field.

The authors addressed my concerns and made substantial helpful revisions based on the comments of the three reviewers, resulting in a stronger paper.

Reviewer #4 (Remarks to the Author):

This is a revised version of a manuscript describing the molecular mechanism by which a small molecule (KNX-002) inhibits the class XIV myosin of Plasmodium parasites. The study is of good technical quality and provides interesting new information on the specific structural mechanism by which KNX-002 binds to post-rigor state and inhibits ATP hydrolysis of Plasmodium myosin A. However, there are few relatively minor points that should be addressed to further strengthen this manuscript.

1. The authors state in the 'Abstract' that KNX-002 targets a novel binding pocket of myosin. This is slightly misleading, because KNX-002 binds to the same binding pocket than blebbistatin, but recognizes a different conformation of the pocket. Thus, it would perhaps be more accurate to e.g. state that KNX-002 displays a novel binding mode to myosin.

2. Introduction to the KNX-002 compound is confusing. The authors should more precisely describe in the 'Introduction' and in the beginning of 'Results' how this compound was identified. Therefore, instead of citing 'Methods', one should cite the bioRxiv pre-print by Kelsen et al., and briefly describe how the compound was identified and what else is known about this compound based on the study by Kelsen et al.

3. In all biochemical assays where $n=2$, it is better to show the individual data points instead of error bars (Fig. 1, Fig. 4, Fig. S1B, Fig. S4C).

4. Figs. 3B and E are quite complex, and need more explanation in the legend.

5. The legend to Fig 4B is confusing, and should be edited.

REVIEWERS' COMMENTS

Reviewer #1 (Remarks to the Author):

As stated in my original review, experiments in the paper are thoughtfully designed, and the paper is well written. The work is impactful and will be of interest to the field.

The authors addressed my concerns and made substantial helpful revisions based on the comments of the three reviewers, resulting in a stronger paper.

We thank Reviewer #1 for his/her positive comments and remarks that allowed us to improve the quality of our manuscript.

Reviewer #4 (Remarks to the Author):

This is a revised version of a manuscript describing the molecular mechanism by which a small molecule (KNX-002) inhibits the class XIV myosin of Plasmodium parasites. The study is of good technical quality and provides interesting new information on the specific structural mechanism by which KNX-002 binds to post-rigor state and inhibits ATP hydrolysis of Plasmodium myosin A. However, there are few relatively minor points that should be addressed to further strengthen this manuscript.

We thank Reviewer #4 for his/her positive comments. His/her remarks were all addressed (see point by point reply below).

1. The authors state in the 'Abstract' that KNX-002 targets a novel binding pocket of myosin. This is slightly misleading, because KNX-002 binds to the same binding pocket than blebbistatin, but recognizes a different conformation of the pocket. Thus, it would perhaps be more accurate to e.g. state that KNX-002 displays a novel binding mode to myosin.

In order to address this point, we edited the manuscript:

“... we demonstrate that KNX-002 inhibits PfMyoA using a previously undescribed binding mode” (Abstract, l. 50-51).

“It is important to describe how the Myo2 inhibitor Blebbistatin (Blebb) and KNX-002 binding modes differ because they involve similar PfMyoA structural elements.” (Results, l. 246-247).

“Taken together, these results clearly demonstrate that KNX-002 binds with a novel and previously undescribed binding mode with unique features.” (Results, l. 263-264).

“KNX-002 thus inhibits PfMyoA using a previously undescribed binding mode requiring residues of the inner pocket in the post-rigor state.” (Discussion, l. 409-411).

2. Introduction to the KNX-002 compound is confusing. The authors should more precisely describe in the 'Introduction' and in the beginning of 'Results' how this compound was identified. Therefore, instead of citing 'Methods', one should cite the bioRxiv pre-print by Kelsen et al., and briefly describe how the compound was identified and what else is known

about this compound based on the study by Kelsen et al.

This was corrected. Some sentences were added:

“KNX-002 was initially identified as an inhibitor from high-throughput actin-activated screens performed by Cytokinetics, Inc.” (Introduction, l. 87-89).

“performed by Cytokinetics, Inc. using 50,000 compounds from their library. The same compounds were screened against TgMyoA in parallel, and upon completion of the screen KNX-002 was identified as a robust inhibitor of both class XIV myosins²³. (see Methods for screen details). KNX-002” (Results, l. 97-100).

3. In all biochemical assays where n=2, it is better to show the individual data points instead of error bars (Fig. 1, Fig. 4, Fig. S1B, Fig. S4C).

This was corrected.

4. Figs. 3B and E are quite complex, and need more explanation in the legend.

The legends of Fig. 3B and 3E were edited:

“(b) Schematic representation of the binding pocket of KNX-002. Each type of interaction is represented differently” (l. 220-221).

Squares indicate residues involved in different types of bonds for KNX-002 and Blebbistatin (shown in (e)). (l. 222-223).

“(e) Schematic representation of interactions around Blebb.” (l. 230).

5. The legend to Fig 4B is confusing, and should be edited.

The legend was fixed and edited as follows:

“(b) When KNX-002 occupies its pocket, the compound stabilizes a water molecule that also binds the γ -P_i of ATP and a water molecule that coordinate the Mg²⁺ ion. **Supplementary Figure 7** indicates that this additional interaction does not change the hexa-coordination of the Mg²⁺ ion.” (l. 326-328)